# How crystallization additives govern halide perovskite grain growth

Timo Maschwitz[1,2,21], Lena Merten [3,4,21], Feray Ünlü[5,6], Martin Majewski [7], Fatemeh Haddadi Barzoki[8], Zijin Wu[9,10], Seren Dilara Öz[11], Cedric Kreusel[1,2], Manuel Theisen[1,2], Pang Wang [1,2], Maximilian Schiffer[1,2], Gianluca Boccarella[1,2], Gregor Marioth[1,2], Henrik Weidner[1,2], Sarah Schultheis[1,2], Tim Schieferstein[1,2], Dawid Gidaszewski[12], Zavkiddin Julliev[13], Ekaterina Kneschaurek [3], Valentin Munteanu[3], Ivan Zaluzhnyy [3], Florian Bertram[14], Anaël Jaffrès[15], Junjie He[1,16], Nigmat Ashurov[13], Martin Stolterfoht [17], Christian M. Wolff [15], Eva Unger [6], Selina Olthof[11], Geert Brocks[9,10,18], Shuxia Tao [9,10], Helen Grüninger [8], Olivier J. J. Ronsin[7], Jens Harting [7], Andreas F. Kotthaus [19] ✉, Stefan F. Kirsch[19], Sanjay Mathur[5], Alexander Hinderhofer [3,20] ✉, Frank Schreiber [3,20], Thomas Riedl [1,2] ✉ & Kai Oliver Brinkmann [1,2] ✉

The preparation of perovskite solar cells from the liquid phase is a cornerstone of their immense potential. However, a clear relationship between the precursor ink and the formation of the resulting perovskite is missing. Established theories, such as heterogeneous nucleation and lead complex colloid formation, often prove unreliable, which has led to an overreliance on heuristics. Most high-performing perovskites use additives to control crystallization. Their role during crystallization is, however, elusive. Here, we provide evidence that typical crystallization additives do not predominantly impact the nucleation phase but rather facilitate coarsening grain growth by increasing ion mobility across grain boundaries. Drawing from the insights of our broad, interdisciplinary study that combines ex and in situ characterization methods, devices, simulations, and density function theory calculation, we propose a concept that proves valid for various additives and perovskite formulations. Moreover, we establish a direct link between additive engineering and perovskite post-processing, offering a unified framework for advancing material design and process engineering.

In recent years, perovskite solar cells have reached power conversion efficiencies (PCE) exceeding 27%[1] along with prospects to achieve years of operational stability[2]. Among other factors, the quality of the active material is most crucial in obtaining a high PCE. A key challenge in the preparation of perovskite thin films from precursor inks is to control the growth of the perovskite grains. During thin film deposition, several processes take place simultaneously, including solvent evaporation, progressing supersaturation, nucleation, and crystal growth[3]. Even minor deviations from an original processing protocol can result in substantial changes in the structure and properties of the deposited material[4]. Therefore, considerable effort has been devoted to control the deposition and crystallization of perovskites[5,6].

A full list of affiliations appears at the end of the paper. ✉e-mail: kotthaus@uni-wuppertal.de; alexander.hinderhofer@uni-tuebingen.de; t.riedl@uni-wuppertal.de; brinkmann@uni-wuppertal.de

Commonly, precursor solvent mixtures, in which at least one solvent is able to form stable intermediate complex structures with the perovskite precursors, are used in combination with solvent extraction methods like anti-solvent-, gas- or vacuum-quenching. These deposition techniques are used to control the morphology of the perovskite layers to reduce the process variables and thereby to ultimately improve reproducibility[7–13]. Larger perovskite grains reduce the density of grain boundaries in the final perovskite layer leading to a reduction of deep trap states and non-radiative recombination—key factors influencing solar cell device performance[14–16]. While there are also other means to mitigate the impact of grain boundaries[17], reduction of the number of grain boundaries by increasing the grain size is the most straightforward and highly popular approach. Aiming to reduce losses and increase perovskite solar cell performance, in addition to the aforementioned varieties of film deposition, various additives may be added to the perovskite precursor ink to impact the resulting perovskite layer. Although the impact of solvent and additive engineering on the final perovskite layer is well documented, the underlying mechanisms governing the crystallization process are still subject of a vigorous debate[18]. In the context of crystallization, the impact of additives is often attributed to a retardation of the nucleation process, caused by a suspected increase in Gibbs free energy due to additive coordination to the lead core[19–25]. A frequently cited hypothesis is a heterogeneous nucleation evolving from $Pb^{2+}$-$MA^+$-$I^-$ clusters that coordinate into $[PbI_6]^{4-}$ octahedra and eventually rearrange into corner-sharing structures that grow in size and finally become seeds for the crystallization of the perovskite phase[26–29]. The existence of colloidal complex structures consisting of the lead component as an acceptor and the polar aprotic solvent as donor molecules, already present in highly diluted perovskite precursor inks, is widely accepted and frequently shown in the literature by multiple methods like UV-Vis, X-ray scattering, and nuclear magnetic resonance (NMR)[13,18,30–33]. Substantial evidence confirms the ability of common Lewis-base solvents to form such complex structures with perovskite precursors[26,34–36]. This concept implies a strong impact of complex coordination on the precursor nucleation and crystallization process, as the size and density of perovskite seeds would determine the size and shape of the grains that are formed in the film. The fact that the lead cation site acts as an electron acceptor (i.e. Lewis-acid) further matches well with several reports stating the beneficial impact of various Lewis-base additives on the crystallization process[20,23,33,37–39]. While those observations are often presented as a chain of causality, a clear systematic link between liquid (ink) phase, nucleation, and grain size is still lacking in the community. So far, nucleation theory is not able to provide solid, generalizable predictions, which is why the community currently relies heavily on intuition and heuristic approaches for process optimization. For process engineering beyond heuristics that will be key for tailored additive design and upscaling, in-depth knowledge of the root cause of perovskite grain morphology is critically needed[40,41].

In this work, we provide a solid concept for the mechanism of how many popular crystallization agents mediate the grain formation in the final perovskite layer and explain why predictions based on nucleation theory are unreliable and prone to error. To this end, we present a multi-facet approach, in which we study the full route of the perovskite formation process, i.e. from probing the precursor ink over in situ grazing incidence X-ray scattering (GIWAXS) during thin film deposition to the final films and their implementation into perovskite solar cells (Fig. 1a). While we are able to confirm and complement reports in the literature on colloidal lead complexes in the perovskite precursor ink that grow in size upon increasing the concentration of the ink, we present compelling evidence that the key impact of several popular additives unfolds during the annealing step, when solvent removal, nucleation and initial perovskite crystallization have already taken place. We model the process of grain growth by phase-field

simulations, using a coarsening growth process limited by the ion mobility of the perovskite constituents across the grain boundaries. The ion mobility in turn, is mediated by the presence of the additive at the grain boundaries of the film. With ultraviolet photoelectron spectroscopy (UPS), density functional theory (DFT) calculation, and Fourier transform infrared spectroscopy (FTIR), we present evidence that coordination of the additive to lead sites, along with the opening of interfacial defect states and/or ion shuttling during annealing, are the underlying mechanisms driving additive mediated grain growth. We validate the generalizability of our hypothesis by testing various perovskite compositions and additives, as well as showing its applicability to complementary post-processing methods. We confirm that the proposed mechanism is not only viable to explain the effect of additives, but also to link the additive mediated perovskite formation to post-processing approaches like thermal hot-pressing, where the mobility of the ions is increased by elevated temperature.

Our findings provide a decisive piece that complements the nucleation theory for perovskite thin film fabrication and bridges the gap between the precursor phase and the final film. Thereby, we take a crucial step beyond heuristics and enable revised and more targeted additive and crystallization engineering.

## Results and discussion

Before we focus on the properties of the perovskite precursor, we briefly address processing parameters that are frequently mentioned to affect the grain size during perovskite film deposition[29,42,43]. Firstly, we deliberately varied the speed of solvent removal (i.e., the evaporation rate, Fig. S1 and Fig. S2), by selecting different deposition and quenching techniques for thin film formation (one step, gas-, and anti-solvent-quenching) with dimethylformamide (DMF) and dimethyl sulfoxide (DMSO), that have different vapor pressures, as the solvents for the perovskite precursor. Secondly, we deposited perovskite films (without any additive) on both hydrophilic and hydrophobic substrates (Fig. S3)[25,29,44,45]. As a proxy for the speed of solvent removal, we recorded the timespan until the perovskite phase formation was visible in the deposited thin films by a distinct color change (Fig. S1). Details on the determination of the grain size in the resulting layers by scanning electron microscopy (SEM) can be found in the Supplementary Note 1. Interestingly, even if the macroscopic morphology may be affected by the drying process, we obtained very similar grain sizes independent of the chosen technique and substrate. We consider the results of this set of initial experiments as a strong indication that our findings for the development of the final perovskite grain size will be transferable to both hydrophilic and hydrophobic substrate types, slow and fast deposition techniques and different popular perovskite solvent systems.

### Probing colloidal lead complexes in the precursor ink

So far, most investigations of lead complexes in perovskite precursors have been conducted on diluted precursor inks using optical absorption techniques or X-ray scattering methods. Due to the highly absorptive nature of the lead-based precursors, optical measurements at high concentrations are experimentally challenging. Our absorption experiments reached a limit at around 0.3–0.5 M (M = mol $l^{-1}$) (see Fig. 1b and Fig. S4). As visualized in Fig. 1c, absorbance allows for the identification of polynuclear lead complexes up to trimers in the precursor solution[30], but in our case proved unpractical for typical perovskite precursor ink concentrations of 1 M or higher.

To probe precursor solutions at higher concentrations, we therefore employed $^{207}Pb$ NMR and electrical conductance (EC) measurement techniques, which are not constrained by the above limitations. NMR is informative about the electronic environment of the lead core, and therefore its complex formation, while EC can be used to probe changes in the volume-to-charge ratio of charged species, which is an indicator for the growth of colloidal aggregates in the solution.

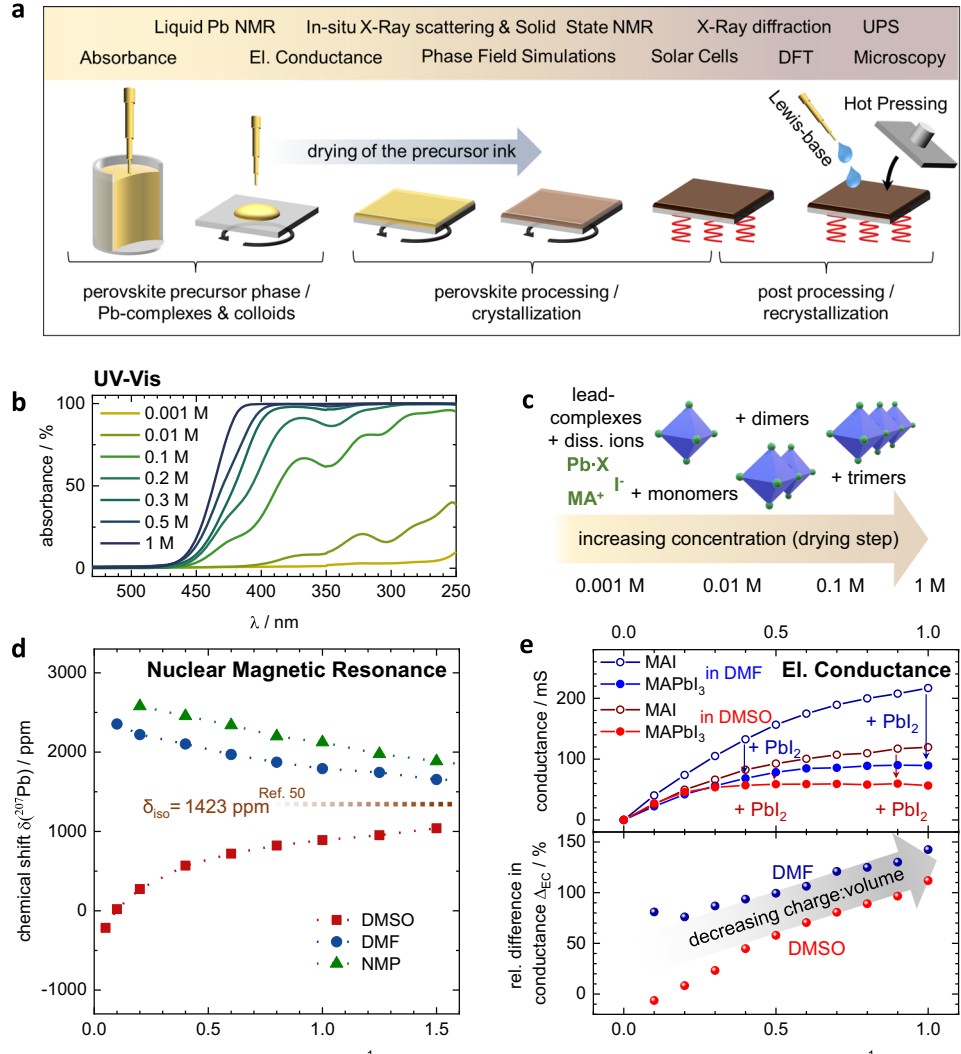

**Fig. 1 | Solvent chemistry—solvent type and concentration. a** Schematic illustration of the perovskite deposition process accompanied by the measurement approach. **b** UV-Vis absorbance measurements on MAPbI₃ precursor inks with different concentrations in DMF. Note that reflection was neglected and absorbance was calculated as $1 - T$, where $T$ is transmittance. UV-Vis for dimethylsulfoxid (DMSO) and n-methyl-2-pyrrilidone (NMP) are shown in Fig. S4. **c** Illustration of the lead complex evolution in dependency of concentration, as also proposed by ref. 30. **d** $^{207}$Pb NMR studies of MAPbI₃ precursor inks based on dimethyl formamide (DMF), DMSO, and NMP in dependency of precursor concentration. The dotted line represents the solid-state MAPbI₃ signal derived from ref. 50. Chemical shifts are reported on a ppm scale relative to the standard Pb(CH₃)₄. **e** Conductance study comparing stoichiometric MAPbI₃ precursor to MAI dissolved in DMF, NMP, or DMSO at different concentration. The bottom panel shows the relative difference of conductance ($G$) between the dissociated MAI and the colloidal MAPbI₃ ink, referenced by MAPbI₃, calculated as $\Delta_{EC} = \frac{G_{MAI} - G_{MAPbI_3}}{G_{MAPbI_3}}$, following the apparent evolution in charge-to-volume ratio in the colloidal ink with increasing concentration. For details about the conductance measurement setup see the Methods section as well as Fig. S5.

In $^{207}$Pb NMR, a high (more positive) chemical shift (referred to as "downfield" shift) indicates a diminishing electron density (de-shielding) around the Pb center while lower chemical shifts (referred to as "upfield" shift) indicate a higher electron density around the lead center. We observed only a single resonance peak from $^{207}$Pb in all inks, because the processes of lead complexation and complex dissociation are fast on the timescale of the $^{207}$Pb NMR measurement and thus the averaged electron density at the lead core is probed[46,47].

To warrant generalizability, we used the three most common perovskite solvents DMF, DMSO, and NMP, all of which are Lewis-bases. DMSO has a slightly higher donor number—a common measure for Lewis basicity—of 29.8 kcal mol⁻¹ in comparison to DMF and NMP with 26.6 kcal mol⁻¹ and 27.3 kcal mol⁻¹, respectively[48,49]. As evident by UV-Vis and NMR spectroscopy (Fig. 1b–d, Fig. S4), all three solvents formed complexes with the lead cation that evolved with precursor ink concentration, in line with earlier findings for diluted precursor inks[13,30,32]. As

expected, due to its higher donor number, DMSO caused an overall higher electron density around the lead nuclei compared to NMP and DMF, as evident by NMR in Fig. 1d. Interestingly, when the concentration was increased, the chemical shifts of DMF and NMP on the one side and the DMSO shift on the other side approached each other on a seemingly asymptotic path. This hypothetical asymptote can be approximated at around 1400 – 1450 ppm and has a striking resemblance with the chemical shifts reported for solid-state MAPbI₃ perovskite (i.e., 1423 ppm)[50]. Considering the averaging effect of $^{207}$Pb NMR, this trend can indicate the shift of the equilibrium of all lead complex species towards the formation of an increasing amount of poly-nuclear, perovskite-like species in the precursor ink, which have been predicted in previous works[13,30].

Similarly, we find indications of growing lead-based complexes by EC measurements, where a comparison of solutions of the MAI salt with the perovskite precursor ink at similar concentrations revealed a reduction in conductance, which we can interpret as a decrease in

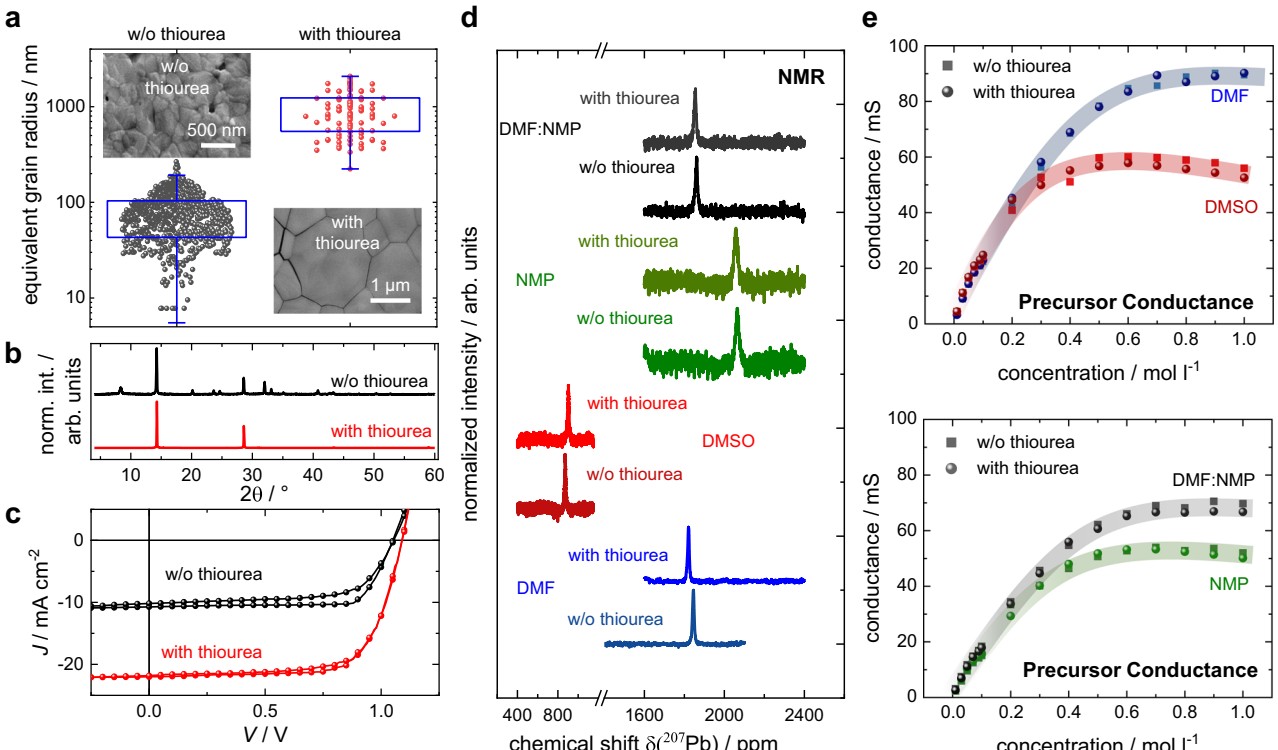

**Fig. 2 | Impact of thiourea additive. a** Grain size distribution and scanning electron microscopy (SEM) images of MAPbI$_3$ layers deposited from an DMF:NMP precursor ink by gas-quenching with a concentration of 1.0 M with and without (w/o) 0.1 M of thiourea additive. Topography recorded by atomic force microscopy and perovskite layers deposited from other solvent systems can be found in Fig. S8. **b** XRD patterns of respective layers with and without 0.1 M thiourea. **c** Solar cell characteristics employing an ITO/PTAA/MAPbI$_3$/PCBM/AZO/Ag device stack using MAPbI$_3$ with and w/o thiourea additive as active layers. Here, measurements in the stabilized state (after light soaking) are shown. Statistics, other solar cell characteristics and solar cells based on a FA$_{0.94}$Cs$_{0.06}$PbI$_3$ active system with and w/o 0.1 M thiourea are shown in Fig. S9, Fig. S10, Fig. S11, and Fig. S12. **d** $^{207}$Pb NMR spectra showing the chemical shift of a 1.0 M MAPbI$_3$ ink with and w/o addition of 0.1 M thiourea in common solvents DMF, DMSO, and NMP, as well as our reference solvent system DMF:NMP. **e** Electrical conductance (EC) measurements of concentration series of the same perovskite and solvent systems as probed by $^{207}$Pb NMR.

charge-to-volume ratio indicating larger and/or less charged species in the ink (Fig. 1e, top). An overall higher conductance for precursors based on DMF in comparison to those based on DMSO can be attributed to different viscosities of the solvents, which directly impact the ion mobility[51–53]. We finally calculated the relative difference between the ink conductances employing either MAI and MAPbI$_3$ in relation to the conductance of the MAPbI$_3$ precursor ink $\Delta_{EC} = \frac{G_{MAI} - G_{MAPbI_3}}{G_{MAPbI_3}}$ (Fig. 1e, bottom). In line with our interpretation of UV-Vis and NMR, the rising $\Delta_{EC}$ with increasing concentration follows the reduction of charge-to-volume ratio, that is consequential from the growth of colloidal structures in the ink. Furthermore, a comparison of DMF and DMSO at lower concentrations, where the $\Delta_{EC}$ differs more than at higher concentrations, agrees well with the asymptotic behaviour of the respective $^{207}$Pb NMR measurements. We speculate that for lower concentrations, the higher donor number of DMSO allows for a stronger dissociation of the PbI$_2$ than DMF, leading to a $\Delta_{EC}$ below zero, meaning that the conductance actually increases upon addition of PbI$_2$. For higher concentrations, in all cases, colloid formation dominates the electronic precursor properties in both solvents. We also found a similar behavior for MAPbI$_3$ in NMP as we show in Fig. S6.

## Influence of additives on grain growth and complex formation

As the variety of additives that have been reported to act as so-called "crystallization agents" is very large, we selected one additive to conduct an in-depth case study and later on verify the general validity of our insights for other additives and perovskite compositions. Initially,

we chose the Lewis-base thiourea, whose effect on perovskite crystallization is well reported and often related to its ability to interact with lead[20,37,38,54,55], e.g. via the formation of hydrogen bonds[56]. Our decision for thiourea as a first case study was based on the fact that it has no secondary interference with the applied measurement techniques. Many other popular crystallization additives like MACl, Pb(SCN)$_2$ or HI impair the complex measurements, as they shift the precursor stoichiometry (Pb(SCN)$_2$, MACl), or add charged species by dissociation (MACl, HI). Altering the precursor stoichiometry generally impacts $^{207}$Pb NMR measurements, independent of the source of the shift (e.g., addition of excess MA, see Fig. S7), while additional ions impact conductance, which makes disentanglement of primary and secondary effects close to impossible in both cases. Thiourea dissolved in DMF, DMSO, or NMP proves to be non-conductive and does not impact the stoichiometry of the perovskite solutions. As we show in Fig. 2a, b, the addition of 0.1 M thiourea to the precursor ink resulted in a drastic increase of the grain size and degree of orientation of perovskite thin films (MAPbI$_3$ in this particular example). We found the impact of thiourea to be largely independent of the solvent-system (Fig. S8) and the perovskite deposition technique (i.e. drying speed, Fig. S9) and therefore chose the DMF:NMP solvent system (7:3 volumetric ratio) and the gas-quenching protocol as a reference deposition procedure which is known for its low process variation[9]. To underline the relevance for applications, we fabricated perovskite solar cells and confirmed a performance improvement due to thiourea addition (Fig. 2c). Further data on the solar cells are summarized in Fig. S10 and Fig. S11. As solar cells without the additive also showed reduced

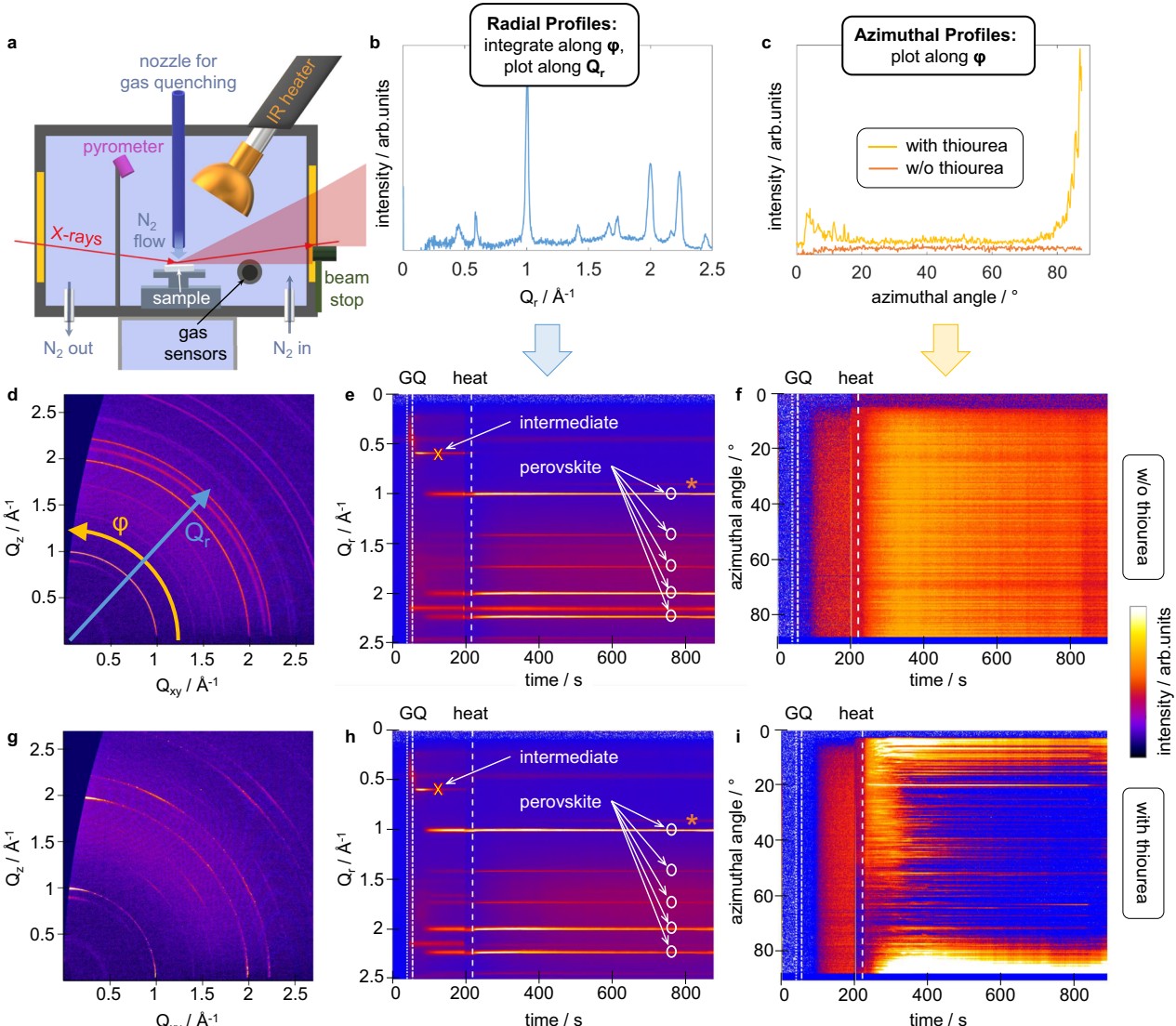

**Fig. 3 | In situ investigations of the deposition process. a** Illustration of the measurement setup for in situ grazing incidence wide angle X-ray scattering (GIWAXS) characterization of the perovskite formation process. Exemplary plots for **b** radial and **c** azimuthal profiles that are plotted versus time in (**e**, **f**), and (**h**, **i**), respectively. Reciprocal space maps of MAPbI$_3$ perovskite thin films deposited from precursors without (**d**) and with 0.1 M (**g**) of thiourea additive. In **d**, also the plot direction for the time-dependent plots is highlighted. Radial Intensities (azimuthally integrated) of reciprocal space maps obtained by in situ GIWAXS as a function of time (**e**, without thiourea, **h**, with thiourea). The decline of the intermediate (cross) and formation of the perovskite phase (circles) in both cases is visible. The orange asterisk marks PbI$_2$. **f**, **i** Azimuthal angular intensity of the perovskite <110> signal (around $Q_r = 1$ Å$^{-1}$) as a function of time, representing orientational order (**f**, without thiourea, **i**, with thiourea). A contrast is visible only after the annealing step has started. Vertical lines in all panels indicate the start of spinning (dotted), gas-quenching (GQ, dot-dashed) and annealing (heat, dashed). For all samples, a solvent mixture of 7:3 DMF:NMP was used.

operation stability, similar as described in earlier work[57], we conducted fast hysteresis measurements (Fig. S12) revealing increased ionic losses in devices without thiourea addition, which are likely related to recombination at the higher number of grain boundaries[16,58]. We found similarly beneficial effects of the thiourea additive for perovskite solar cells with a FA$_{0.94}$Cs$_{0.06}$PbI$_3$ composition (Fig. S13) as well as mixed halide perovskites, which we published earlier[59,60].

Despite such dramatic changes in the perovskite layers upon addition of thiourea, we could only detect minimal variations in the electronic environment of the lead nuclei by NMR spectroscopy, which are within the error margin of the highly sensitive $^{207}$Pb NMR measurement (Fig. 2d). Likewise, the impact of thiourea on the conductance of the respective inks proved to be negligible (Fig. 2e). Taken together, we conclude that if thiourea were to influence the complex and colloid formation in the precursor ink, it would be below the detection limits of our measurements. This finding is unexpected,

because thiourea, as a sulfur-donor, has been predicted to possess a higher coordination affinity towards lead compared to oxygen donors such as DMF, NMP and DMSO[34,61,62]. Based on our combined data, we suspect that the decisive impact of the crystallization agent on the perovskite film morphology may occur later in the deposition process.

### In situ GIWAXS with and without additive
In an effort to access the later stages of the perovskite deposition process (supersaturation, crystallization, and annealing), we constructed a setup that enables us to monitor the perovskite formation during spin-coating and different quenching processes by recording in situ GIWAXS data using synchrotron X-ray radiation (Fig. 3a)[63,64]. A comparison of perovskite layers annealed via infrared annealing or on a hotplate can be found in Fig. S14. Besides the apparent increase of grain size when employing the thiourea additive, also the texture of the perovskite film was strongly affected (Fig. S15). While for perovskite

layers processed without the thiourea additive we found an essentially random orientation of the crystalline domains, layers processed with thiourea showed a distinct orientational order with the < 110 > direction perpendicular to the substrate surface. For coherently scattering crystallite domains with more than a few hundred nanometers in size, precise dimensions cannot be extracted reliably from the peak width as this is limited by resolution effects[65,66]. Thus, for the thiourea additive we chose to use the distinct contrast in crystallite orientation as an additional indicator to monitor the dynamics of the grain formation process which is established simultaneously with the size of crystallites and, ultimately, grains. Along with the radial profiles of the measured reciprocal space maps (integration along the azimuthal angle $\varphi$, plot along the radial direction $Q_r$, Fig. 3b), we plot the azimuthal profiles of the MAPbI$_3$ < 110 > signal at $Q_r = 1$ Å$^{-1}$ (plot along $\varphi$, Fig. 3c)[67]. The corresponding time-resolved GIWAXS data is shown in Fig. 3d-i. The full reciprocal space maps at distinct points in time during perovskite formation can be found in Fig. S16 and S17. We want to note that a dynamic system of perovskite-like colloidal structures, where complexes continuously form and dissociate, differs substantially from a nanoparticle dispersion and does not imply crystallinity, which is in line with our observation by GIWAXS, showing no detectable crystalline features in the precursor ink (Fig. 3e, h, before the spin-coating is started). We speculate that these dynamic processes are also the reason why dynamic light scattering techniques, which use models of dispersed particles, are unreliable when probing perovskite precursors[68,69].

The first crystalline feature can be seen briefly after the gas-quenching started in both samples with and without thiourea. The signal at $Q_r = 0.6$ Å$^{-1}$ (crosses, Fig. 3e, h) indicates the PbI$_2 \cdot$ NMP solvent complex structure, which has been reported to occur as an intermediate phase in perovskite formation (Fig. S18)[70,71]. The lower vapor pressure of NMP and its greater coordination affinity to the lead core compared to DMF[34] renders it more likely to remain in the film during spin-coating and gas-quenching. Interestingly, as indicated by the emergence of the signal around $Q_r = 1$ Å$^{-1}$ (circles, Fig. 3e, h), a substantial conversion of this intermediate structure into a MAPbI$_3$ perovskite phase takes place already during the gas-quenching process[67]. Azimuthal profiles reveal random orientation of the perovskite phase crystallites formed during the spin-coating and gas-quenching, regardless of whether thiourea was added or not (Fig. 3f, i; before dashed line). Upon annealing (after the dashed line), the situation changes completely, as the perovskite layer with thiourea undergoes a drastic reorientation, resulting in a thin film with highly oriented crystallites as the end product, while its counterpart without thiourea remains randomly oriented.

To illustrate the process, we plotted the intensity of the intermediate and the perovskite phase along with an indicator for the degree of order of the perovskite, represented by the quotient of scattering intensities from the perovskite phase at 90° and 45° azimuthal angle from Fig. 3f, i. Thereby, a representation of the sequence of processes during perovskite layer formation can be drawn, as shown in Fig. 4. On the one hand, the development of the intermediate and perovskite phase (top panels of Fig. 4b) clearly shows similarity in the initial crystallization kinetics with and without thiourea throughout the supersaturation, nucleation, intermediate phase conversion and the perovskite formation process. On the other hand, the degree of orientational order (bottom) reveals that the thiourea additive reorients the perovskite phase upon annealing. In line with established literature, we found a coincidence of large grain size and crystallite orientation along the face lattice planes (for MAPbI$_3$ in < 110 > direction)[72,73] and therefore conclude that the increased grain size, caused by the influence of thiourea, emerges concurrently during the annealing step.

We elucidated the final disposition of the thiourea additive by Fourier transform infrared (FTIR) spectroscopy. As shown in Fig. S19,

the thiourea molecule was detected inside the annealed perovskite film along with a distinct wavenumber shift of the C=S stretching compared to the bare material, which points towards a coordination between lead and thiourea in the solid film, as we found a similar shift when thiourea is added to PbI$_2$, but not if added to MAI. Taken together, our observations can be condensed to the sequence shown in Fig. 4c: Colloidal lead complexes in the precursor grow in size upon spin-coating and concomitant increase of concentration. During gas-quenching, supersaturation, nucleation and formation of the initial perovskite occur and, eventually, during the annealing step, the additive thiourea causes a reorientation and grain growth of the perovskite, while the grains in layers without additive remain smaller and randomly oriented.

## Generalizable coarsening approach for perovskite grain growth

Our findings contradict the hypothesis that upon addition of thiourea, retarded nucleation and crystallization would be the origin of the increased grain size. As a first validation, we investigated the consistency of our findings at different precursor concentrations, where we found a general dependency of the initial perovskite grain size of the pristine materials on the grain aspect ratio (i.e., grain equivalent radius vs. film thickness, see Fig. S20). This is in line with the inherent dependency of grain coarsening on the aspect ratio of the grains which depends on the layer thickness. The addition of thiourea nevertheless resulted in an increase in grain size of almost an order of magnitude in all cases. To furthermore test the validity of our findings obtained with thiourea for other additives and perovskite compositions, two additional sets of in situ GIWAXS studies were conducted: firstly, we replaced the sulfur donor thiourea by the oxygen donor additive urea and secondly, we exchanged the MA$^+$ on the A-site of the perovskite to obtain FA$_{0.94}$Cs$_{0.06}$PbI$_3$. In both cases, we observed a clear increase in grain size with the additive (SEM images in Fig. 5a, b). In contrast to the case of MAPbI$_3$ with thiourea, in the other cases the grain size increase is not accompanied by a strong orientation of the perovskite, which is why we resorted to the fluctuation of the signal intensity along the azimuthal angle as an indirect measure for the grain size during the in situ measurement (see Fig. S21 and Supplementary Note 2 for an in-depth discussion of the choice of metric). As shown in Fig. 5a, for all cases the observations were rather similar to the thin film of MAPbI$_3$ with thiourea. On the one hand, the impact of the additive was minimal during the initial formation process of the perovskite phase (or an intermediate $\delta$-phase in case of FA$_{0.94}$Cs$_{0.06}$PbI$_3$) and the formation of colloidal complex structures in the precursor solution (Fig. 5a, for precursor ink data see Figs. S13 and S22). On the other hand, including the additives caused a drastic increase in azimuthal signal fluctuation during annealing, which indicates growth of the large grains, when an additive is employed, compared to a much smaller increase in a case without additive.

Based on the previous experiments, which showed that grain growth occurs after the perovskite phase already has been developed, we anticipated that a similar process might also take place on a fully fabricated, crystallized perovskite film. Therefore, we tested post-treating a pristine MAPbI$_3$ perovskite film (processed without additive) with either thiourea or urea dissolved in isopropanol followed by annealing at 100 °C. As shown in Fig. 5b and Fig. S23, post-treatment can yield almost identical results to the case where the additive had been added to the precursor solution. These results motivated us to extend the investigation to the widely used additive MACl, which also has shown promise for post-treatment,[74,75] as well as another group of additives based on pseudo-halides[37,76–78], where we chose ammonium thiocyanate, methylammonium cyanate, and methylammonium thiocyanate for their ability to be dissolved in an orthogonal solvent, as shown in Fig. S24. Previous studies have often attributed grain size enhancement in the context of MACl or pseudo-halides to nucleation effects[78]. However, our findings demonstrate

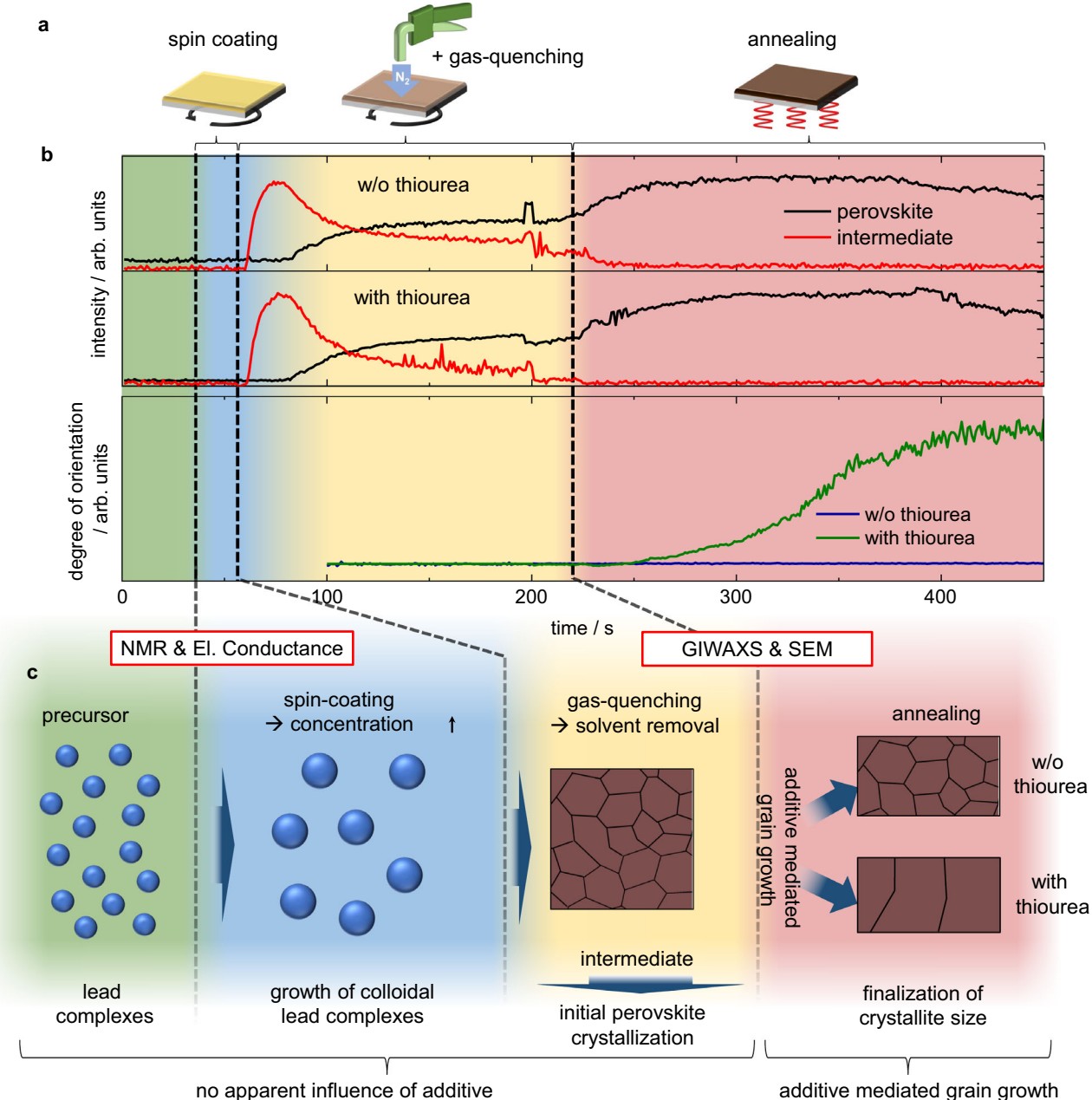

**Fig. 4 | Perovskite formation process. a** Schematic illustration of the deposition process during the in situ experiment. **b** (top) Integrated GIWAXS signal intensities of perovskite and $PbI_2 \cdot NMP$ intermediate as a function of time without and with thiourea. The intensities that correspond to the respective phases are indicative of the relative material portions. (bottom) Degree of orientational order, represented by the ratio of the integrated GIWAXS signal intensity at 90° against the intensity at 45° azimuthal angle. Orientational order is used as an indicator for the formation of large grains with preferentially oriented crystallites as ultimately observed in layers employing the thiourea additive. For all samples, a solvent mixture of 7:3 DMF:NMP was used. Artefacts at around 200 s are related to the stop of the spin-coating process. **c** Sequence of processes as derived from precursor analysis (UV-Vis absorption, NMR & electrical conductance) and layer formation studies (GIWAXS & SEM), both in situ and ex situ. In the perovskite precursor, polynuclear colloidal structures form and increase in size during the spin-coating and drying process. Rapid solvent evaporation during gas-quenching leads to the formation of the $PbI_2 \cdot NMP$ intermediate, which transitions into the perovskite phase with continued nitrogen flow. During the annealing step, the last residues of $PbI_2 \cdot NMP$ intermediate are eliminated and the impact of the additive unfolds: In the absence of the additive, the randomly oriented perovskite phase with small grains is retained, while in the presence of the thiourea additive, highly oriented and large crystallites on the micrometer scale emerge. Coordination of the thiourea additive to the lead site likely occurs shortly before or during the annealing step.

that similar increases in grain size occur regardless of whether the additives are incorporated during thin film deposition or introduced via post-processing. Validation with $FA_{0.94}Cs_{0.06}PbI_3$ as the perovskite composition, while using multiple additives, exhibited a similar trend as shown in Fig. S25. Taken together, our findings challenge the applicability of the currently presented nucleation-based explanations, clearly rendering them insufficient to account for the grain growth effects observed with these additives and

underscore the universality of the coarsening mechanisms for additive mediated grain growth that we describe later.

The distinct grain growth mediated by post treatment with thiourea, urea, or MACl including subsequent annealing also shows some resemblance to planar hot-pressing of perovskite layers. The latter uses thermal imprint, where re-organization of the perovskite material under the concomitant impact of pressure and heat yields large and highly oriented grains from originally small and randomly

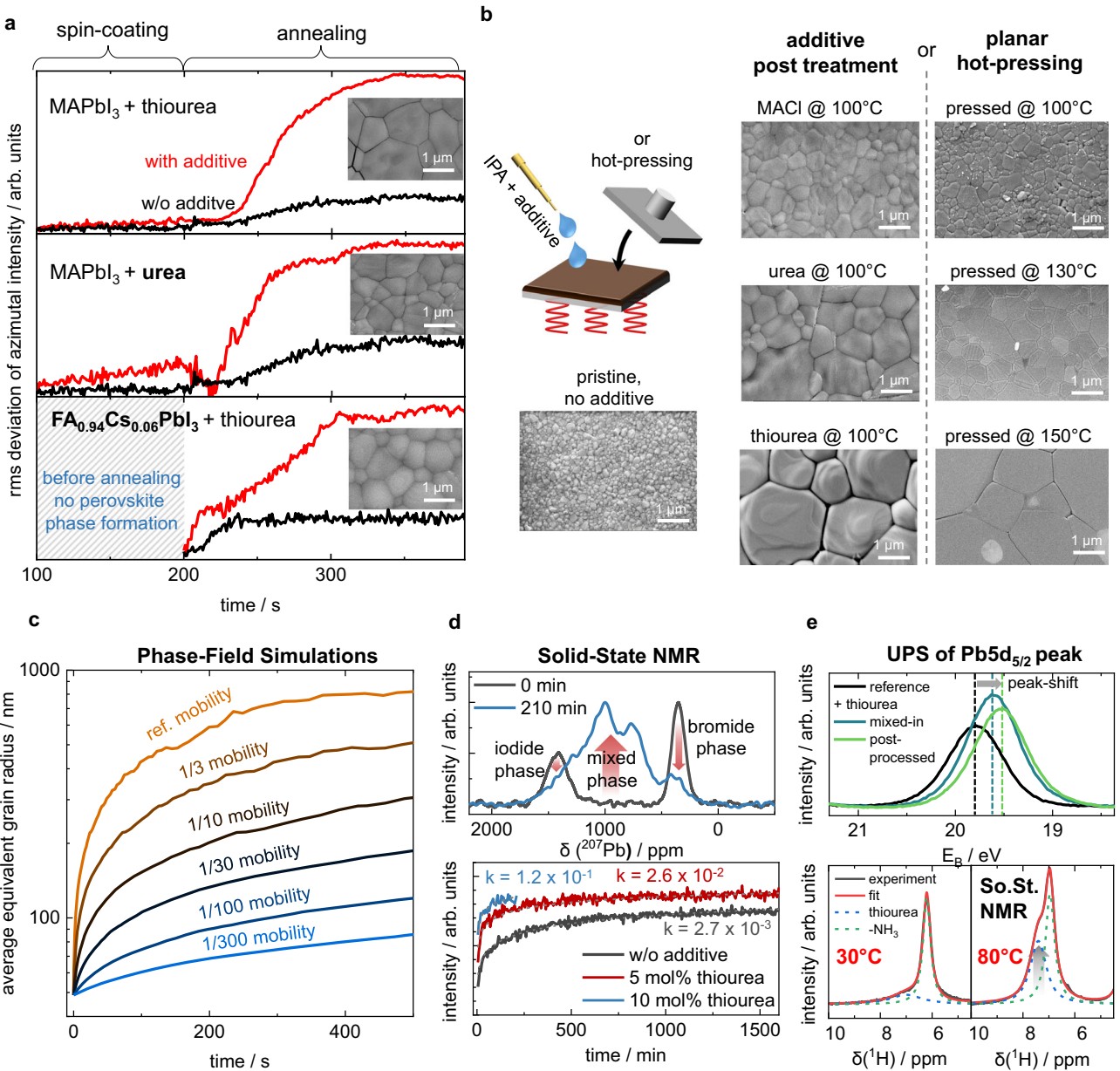

**Fig. 5 | Universal coarsening mediated growth of perovskite grains. a** Evaluated GIWAXS measurements plotting the root-mean-square (rms) of azimuthal intensity fluctuations of characteristic diffraction signals as a function of time, obtained from time-resolved GIWAXS measurements, providing an indirect measure of the grain size. The plots show MAPbI3 with and without 0.1 M thiourea and 0.1 M urea additive, as well as FA0.94Cs0.06PbI3 with and without 0.1 M thiourea additive. The insets show SEM images of the respective perovskite layers. **b** Illustration and results of the post-processing approaches, where the crystallization agent was either spin-coated from isopropanol (IPA) solution on top of the pristine layer with subsequent heating (see also Fig. S23) or planar hot-pressing was applied at different temperatures. **c** Phase-field simulations illustrating a coarsening grain growth with different grain boundary mobilities as a function of time. For details, see Supplementary Note 3. **d** In situ ion mixing experiment observed by solid-state NMR to track the ion transport kinetics with and without the thiourea additive. (top) Mixing of MAPbI3 and MAPbBr3 without additive before and after 210 min

annealing at 80 °C. (bottom) Normalized NMR signal intensities of the [PbI6−xBrx] environment ("mixed phase") during the mixed-halide perovskite phase formation process from physical mixtures of MAPbI3 and MAPbBr3 containing 0 mol-%, 5 mol-%, and 10 mol-% thiourea fitted by Johnson-Mehl-Avrami-Kolmogorov model; the parameter $k$ is reported for quantification. Fitting parameters and further details are given in Table S2 and Supplementary Note 4. The red arrows illustrate the NMR signals that are tracked for the bottom panel. A similar study for MACl additive is shown in Fig. S33. **e** (Top) Ultraviolet photoelectron spectroscopy (UPS) measurements of MAPbI3 with and w/o thiourea, showing a shift of the Pb 5d5/2 semi-core level to lower binging energy in the presence of thiourea, indicating electron transfer from the additive to the lead atoms at the sample surface. The full valence band scans and data for MACl treatment are included in Fig. S34. (Bottom) Temperature-dependent solid-state NMR measurements focusing on the ¹H signal of NH2 groups (-7 ppm). The grey arrow indicates the position and development of the thiourea signal.

oriented perovskite grains[79,80]. Indeed, a temperature series for planar hot-pressing showed that with temperatures of 150 °C, similar grain sizes can be achieved as with the thiourea additive and annealing at 100 °C (Fig. 5b). A second hot-pressing experiment, where the MAPbI3 already contained the thiourea additive in the pristine layer, revealed

that including the thiourea additive during imprinting even furthers the grain growth (Fig. S26).

Considering all the observations from above, we propose the following path for additive mediated perovskite grain formation: Instead of retarding nucleation and crystal formation from solution,

we postulate that most additives might spark their impact during a secondary grain growth process (also sometimes referred to as "grain coarsening"). Grain growth in a polycrystalline material is generally driven by interfacial tension as it leads to the minimization of the energy associated with grain boundaries. Large grains grow at the expense of the smaller ones, similar to the Ostwald ripening process. The coarsening rate increases not only with the grain boundary energy, but also with the mobility of the grain boundaries[81,82], which represents the dynamics of material exchange amongst grains. In this context, additives and/or temperature can mediate the transport of the ionic perovskite components across grain boundaries and therefore increase the mobility of the grain boundary.

We performed phase-field simulations to model the mobility-dependent grain growth. Additionally, as a complement to the GIWAXS data, we performed a dedicated SEM series taken at different stages of the coarsening process to indicate the general kinetics of the grain growth. As evident in this series in Fig. S27 as well as in GIWAXS Fig. 5a, a substantial slowing of the grain growth occurred some min after the start of the annealing. The apparent saturation in the grain size is in contrast to the theory of "normal" grain coarsening, where the average grain size grows with the square-root of time[83]. In our phase-field simulations, we therefore additionally considered that the mobility will probably vary between grain boundaries. A statistical variation of grain boundary mobilities is a known source of grain growth stagnation[84]. An in-depth discussion on the topic of stagnation can be found in Supplementary Note 3. Figure 5c clearly highlights the impact of the grain boundary mobility on the crystal size during grain coarsening. While such a behavior might feel intuitive for a hot-pressing process, where the grain boundary mobility can be expected to follow an Arrhenius-type of temperature behavior[85], assuming a similar effect for crystallization additives might be less obvious. In light of our combined experimental and theoretical findings, we anyhow conclude that the root cause for the outcome of additive mediated perovskite crystallization is an increased ion mobility that accelerates the grain coarsening process, while the impact of the nucleation and crystallization kinetics is negligible.

To further verify this hypothesis, we conducted a dedicated solid-state NMR study to probe the halide kinetics, which we use as an in situ observable proxy for the ion mobility across grains[86]. Specifically, we mixed and heated (80 °C) MAPbI$_3$ and MAPbBr$_3$ powders with and without thiourea and simultaneously tracked the formation of the mixed MAPb(I$_{0.5}$Br$_{0.5}$)$_3$ by following the evolution of the $^{207}$Pb environment of [PbI$_{6-x}$Br$_x$] in $^{207}$Pb magic-angle-spinning (MAS) NMR spectroscopy (for details see Supplementary Note 4, Figs. S31 and S32). As we show in Fig. 5d, the formation kinetics of [PbI$_{6-x}$Br$_x$] signals for perovskite powders with thiourea are significantly faster compared to the case without thiourea, confirming that thiourea significantly accelerates the halide migration kinetics across grain boundaries. A similar solid-state NMR halide mixing study for the MACl additive also revealed increased ion mobility in the presence of the additive, albeit far less pronounced (Fig. S33).

**Origin of the increased ion mobility in the presence of additives**
To understand the underlying mechanism by which the additive interacts with the perovskite and thereby mediates the increase in ion mobility, we employed additional measurements by ultraviolet photoelectron spectroscopy (UPS, Fig. 5e, top), temperature-dependent $^1$H solid-state NMR (Fig. 5e, bottom) and also carried out extended density functional theory (DFT) calculations. Together, these results allow us to propose a hypothesis describing how additives promote grain coarsening by interacting with grain boundaries to enhance ion mobility, as illustrated in Fig. 6. Note that, while we observed a similar overall trend, we found slightly smaller final grain sizes of FA$_{0.94}$Cs$_{0.06}$PbI$_3$ compared to MAPbI$_3$ upon thiourea addition. This difference could be related to inherently

different ion mobilities within the perovskite grains, as also suspected in the context of halide segregation, where FA-based perovskites also show a lower tendency to halide-segregate[87].

To investigate the possibility of charge transfer between the additive and the perovskite at the grain boundary, we performed high-energy resolution UPS measurements using a monochromatic He II$\alpha$ excitation at an energy of 40.8 eV. This allows to monitor the binding energy of the Pb 5d$_{5/2}$ semi-core levels close to the surface. The comparison of the Pb 5d$_{5/2}$ peak of a reference sample, as well as films either incorporating thiourea or being post-treated by it, are shown in Fig. 5e (full spectra can be found in Fig. S34). Here, thiourea leads to a shift of the Pb semi-core level toward lower binding energies, indicating electron donation to the lead atoms. This observation aligns with our FTIR data (Fig. S19), allowing us to conclude that thiourea is coordinated to lead sites at room temperature. While in a bare perovskite material it is expected that methylammonium (MA) terminates the surface[88], the UPS measurement indicated that MA is replaced on the surface by the lead-coordinated thiourea (Fig. 6d), regardless whether the additive was mixed into the precursor or added post-process. Please note that, as we cannot determine the surface coverage of thiourea from our experiments, it is possible that the surface termination is inhomogeneous, comprising a mixture of terminal groups.

To obtain a deeper understanding of the underlying mechanisms that further the ion migration, we then pursued an extensive DFT study (Supplementary Note 5, Tables S4, S5, Figs. S36–S38). Our DFT calculations reveal that coordination to lead is energetically favorable via the sulfur atom of thiourea, which aligns well with our observations with both FTIR (Fig. S19) and UPS (Fig. S34). Calculations of the adsorption energies comparing additives and solvents (Table S5), also reveal the strong coordination of NMP with the perovskite. As such it is a likely scenario that thiourea impact occures, in line with our GIWAXS observations, in the annealing step, where remaining strongly coordinating solvent molecules like NMP are expelled from the perovskite crystal surface and the additives are able to take their place.

At the same time, at elevated temperatures, solid-state $^1$H NMR spectroscopy (Fig. 5c, bottom, Fig. S33) shows a narrowing of the $^1$H resonance at around 7.5 ppm corresponding to the NH$_2$ groups of thiourea (decreased FWHM), indicating greater molecular mobility and suggesting that thiourea also readily detaches from the perovskite surface. This detachment gives rise to two key effects−defect opening and ion shuttling−that both can enhance ion migration, as shown in Fig. 6e. Upon detachment, previously passivated defect sites become exposed. These unpassivated defects or vacancies are known to serve as channels for ion migration, thereby facilitating enhanced ion mobility[89,90]. Note that in the context of halide mixing, we previously observed increased ion mobilities in the presence of ionic liquids[86]. In the context of ion shuttling, DFT calculations of bonding energies (Table S4) and electrostatic potential distributions (Fig. S37) further reveal that both urea and thiourea exhibit strong coordination between NH$_2$ groups and iodide, making them potential iodide shuttles that enhance ion transport across grain boundaries. Furthermore, thiourea has a significantly lower detachment energy compared to urea (i.e. a lower adsorption energy, see Table S5), which is consistent with our experimental observation of a more pronounced grain coarsening effect in the presence of thiourea (Fig. 5b).

In contrast to urea and thiourea, MACl lacks the ability to function as an ion shuttle. In this case, the primary mechanism enhancing ion migration remains the opening of defects upon MACl expulsion during annealing[91]. This results in a mobility ($\mu$) sequence of $\mu_{thiourea} > \mu_{urea} > \mu_{MACl}$ (Fig. 6c), which aligns well with the observed grain sizes (Fig. 5b), where a higher ion mobility correlates with higher coarsening rates and concomitantly larger grain sizes. To validate our hypothesis, we extended our solid-state NMR study of halide mixtures (Fig. 5d) by performing an analogous experiment

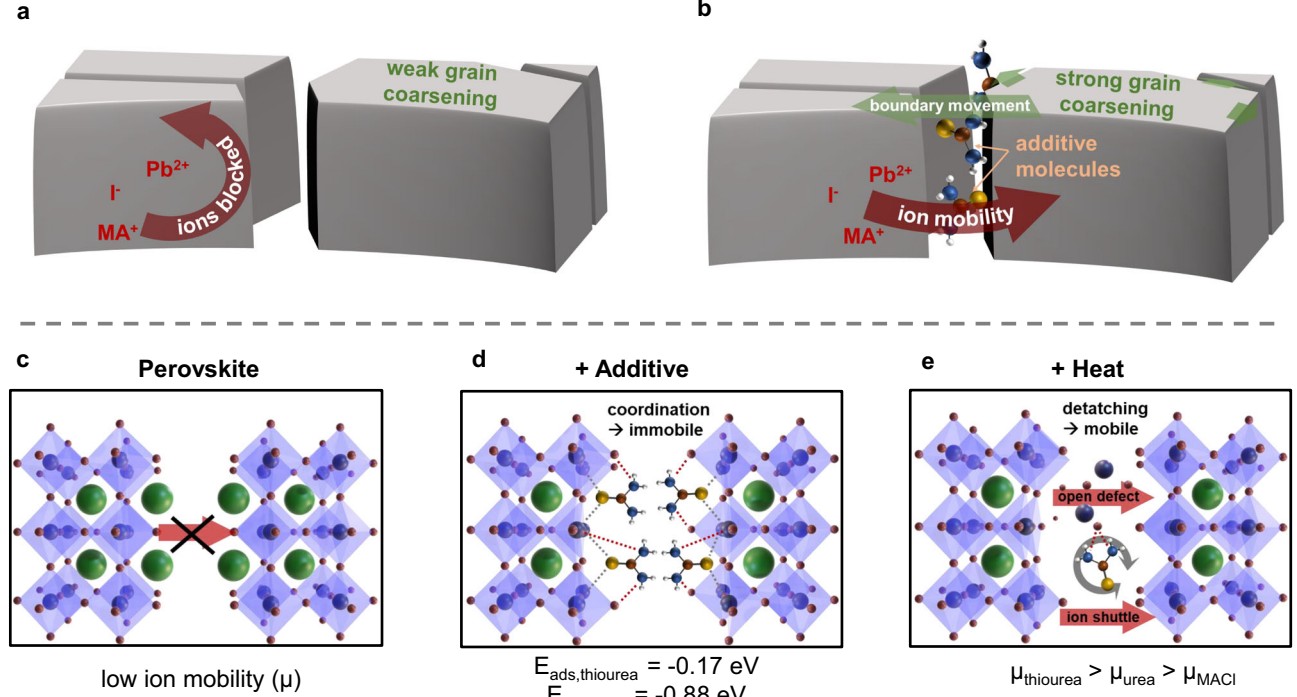

**Fig. 6 | Atomistic mechanism of how additives can increase ion mobility and grain coarsening.** Schematic illustration of the coarsening effect **a** without and **b** with the presence of additives at the grain boundary that enhance ion mobility. **c**–**e** Mechanisms of enhanced ion mobility in the presence of additives and elevated temperatures, as derived from FTIR, UPS, solid-state NMR measurements, and DFT calculations. Both processes, defect opening and ion shuttling, appear to increase ion mobility and further grain coarsening, which is why urea and thiourea, which can act as ion shuttles, generate higher ion mobilities than MACl, which is only expected to open defects. Furthermore, the lower adsorption energy of thiourea in comparison to urea makes it more likely to detach and gain mobility, as also indicated in Fig. 5e at elevated temperatures.

with MACl as the additive as shown in Fig. S33. The results confirmed our hypothesis, showing a five-fold increase in ion mobility for MACl —significant, but still far less than the ~50-fold increase observed for thiourea. We further found a combination of the two additives MACl and thiourea to be substitutive (i.e. the additive that impacts ion mobility the most defines the final appearance of the grains as shown in Fig. S39), which is well in line with our hypothesis, as the amount of possible ion channels is limited by the amount of interface defect states available.

Ultimately, we are therefore able to confirm our hypothesis for perovskite grain growth that is apparently not only widely applicable for additive engineering but also connects a plethora of different grain-manipulation approaches used in the perovskite community. Other approaches, such as the large variety of vapor exposure techniques (solvent, (di-)methylamine, moisture,...)[92–94] are also readily explicable by mobility-driven coarsening, as partially speculated earlier[95]. Given the general applicability of our hypothesis, we reckon that advancing beyond the nucleation and deliberately targeting and designing additives to mediate ionic mobility, will be key to finally move past the "trial-and-error" approaches and enable actual model-driven process engineering for perovskite film deposition, which is critically needed for successful and economic upscaling of the technology.

## Discussion

In conclusion, we present compelling evidence that challenges the widespread conception of perovskite grain growth and ultimately also the quality of a perovskite layer in the presence of crystallization additives to be vastly governed by the nucleation process. We conducted a broad interdisciplinary study of perovskite film formation from precursor inks with and without additives, where we covered the entire range from precursor chemistry over in situ investigation of thin film formation to device integration. By a combination of solid and liquid state NMR, electrical conductance measurements in the precursor ink, in situ and ex situ GIWAXS, XRD, and FTIR measurements encompassed by theoretical modeling and comparison to post-processing approaches, we could find evidence that additive engineering and most of the post-process treatments follow the same set of rules. While we were able to monitor the evolution of lead complexes and colloidal development, we could not find any consistent connection that would link the colloid formation in the precursor to the final morphology of the perovskite thin film. Instead of targeting (i.e. retarding) the nucleation dynamics, the key feature of successful crystallization engineering seems to be the ability of an additive, or a post-processing strategy, to speed up ion transport across perovskite grain boundaries. With a combined effort of DFT calculations, temperature-dependent solid-state NMR, and UPS, we found evidence that the additives enable a coarsening-driven grain growth, after the nucleation has already taken place, either by opening up defect sites or by additionally acting as an ion shuttle upon annealing. Our results appear to be universal for various popular crystallization additives with iodide-containing lead-perovskite materials, which include most of the solar cell materials currently in use. We are convinced that re-evaluating the community's working hypothesis on perovskite formation will enable the design of targeted crystallization approaches and ultimately set a milestone on the road to a model-driven process engineering for perovskite thin film deposition.

## Methods

### Preparation and fabrication

**Materials, layer, and device fabrication.** Extra dry solvents were purchased from Sigma-Aldrich (anhydrous, DMF 99.8%, NMP 99.5%, DMSO 99.9%, toluene, 99.8% purity, isopropanol, 99.5% purity) and

Acros (chlorobenzene, 99.8%). Lead iodide was purchased from Apha Aesar (ultra-dry, 99.999% purity), and organic salts from Greatcell Solar (purity > 99.99%). Thiourea was purchased from Alfa Aesar (99% purity). PTAA (99.9% purity) was purchased from Sigma-Aldrich. MeO-2PACz (purity > 98%) and PEAI (purity > 98%) were acquired from TCI. PCBM (purity > 99%) was purchased from Ossila Ltd. AZO nanoparticles (12 nm, aluminum-doped zinc oxide, mixture in alcohols) with the product code N-21x are commercially available from Avantama AG.

**Perovskite ink preparation.** Perovskite inks were prepared stochiometric either in pure solvents (precursor studies and layers) or in a 7:3 mixture of DMF and NMP. When needed, thiourea additive was dosed from a 100 mg ml$^{-1}$ stock ink in the respective solvent (DMF in case of mixed solvent precursors) before processing or measurement. The amount of solvent was always chosen in a way to warrant the correct concentration of the perovskite precursor salts. For dissolution, all precursors were stirred for at least 4 hours at 60 °C. Before processing, the perovskite ink was filtered through a 0.2 μm PTFE-filter.

**Perovskite layer deposition.** If not stated differently, the perovskite deposition was conducted via the gas-quenching approach as published in our previous report[9]. Briefly, 180 μl of the perovskite precursor was dispensed onto an ITO-coated glass substrate and spin-coated at 3000 rpm (2000 rpm for devices to increase current density output) with a ramp of 4 s and a hold time of 120 s. After 20 s (3 s for pure DMF) a nitrogen flow aiming at the substrate (7 bars, nozzle distance ~2–3 cm) was switched on and maintained until the end of the spin-coating process. For the deposition route using the anti-solvent technique, 600 μl of chlorobenzene was dropped at the same moment of time as selected for the start of the gas-quenching. For one-step processing the hold time was increased to 360 s (DMF) or 540 s (DMSO). For drop casting, 20 μl of the perovskite ink was dispensed onto the substrate and the liquid was left on the substrate until it had dried. After spin-coating or drop casting, the substrate was annealed for 10 min (20 min for FA$_{0.94}$Cs$_{0.06}$PbI$_3$) at 100 °C on a hot plate under an inert atmosphere. The investigations with GIWAXS made use of an infrared radiation source to achieve the annealing step. For FA$_{0.94}$Cs$_{0.06}$PbI$_3$ solar cells, afterwards a 2.5 mg ml$^{-1}$ solution of PEAI in IPA was spin-coated at 6000 rpm for 20 s and subsequently annealed for 10 min at 100 °C to form a 2D-capping layer.

**Perovskite layer post-processing.** For post-processing, solutions of thiourea, urea, and MACl with concentrations of 10 mg ml$^{-1}$ were used, which have been prepared the same way as 100 mg ml$^{-1}$ thiourea solution. 20 μl were dynamically spin-coated on top of the already fabricated and annealed perovskite layer at 3000 rpm and left rotating for another 10 s. Afterwards, the layer was annealed at 100 °C for 10 min.

**Planar hot-pressing.** Planar hot-pressing was conducted in a press system using parallel steel-plates, which we used frequently in previous work for imprinting of various perovskite materials[79,80,96]. The steel plates can be electrically heated and cooled down by water. The samples were packed in N$_2$-atmosphere in a glovebox together with a silicon stamp (dimensions 15 × 15 mm$^2$) which had been coated with an anti-sticking layer. For increasing the heat transfer to the sample, rubber-based thermally conductive foil (KU-CG20/R, Boyd) was used on top and below the sample-stamp stack. After insertion to the press setup, the pressing plate surfaces approached each other, and the pressure was gradually increased up to 100 bar. Heating was then turned on at the desired temperatures and held for 10 min. Afterwards, cooling was applied, and the pressure released as soon as a temperature of 25 °C was reached.

**Solar cell fabrication.** For solar cell fabrication, an excess of 5% PbI$_2$ was added to the precursor for defect passivation, and a substrate structured with a photoresist to define the active area of the solar cells to 3.14 mm$^2$ was used. To warrant homogeneous coverage also on the photoresist, the acceleration ramp of the spin-coater was set to 10 s. The solar cell stack was glass/ITO/HTL/Perovskite/PCBM/AZO/Ag. HTL was PTAA for MAPbI$_3$ or MeO$_2$PACz for FA$_{0.94}$Cs$_{0.06}$PbI$_3$. PTAA was solvated in toluene with a concentration of 5 mg ml$^{-1}$, stirred overnight at 60 °C, and filtered through a 0.2 μm PTFE filter. Then, 120 μl of PTAA solution was dispensed on the substrate and spin-coated at 6000 rpm for 20 s, followed by a 10-minute annealing step at 120 °C. PCBM (25 mg ml$^{-1}$) was dissolved in chlorobenzene by stirring overnight at 115 °C and was filtered through a 0.2 μm PTFE filter after cooling to room temperature. 80 μl of PCBM solution was spin-coated on top of the perovskite at 1000 rpm for 60 s. AZO was diluted in isopropanol with a 1:2 volume ratio and 100 μl were spin-coated at 4000 rpm for 20 s followed by an annealing step at 80 °C for 70 min. As an electrode, 100 nm of silver was thermally evaporated in high vacuum.

**Synthesis of perovskite powders for solid-state NMR.** Perovskite powders of MAPbI$_3$ and MAPbBr$_3$, both with and without 5 mol-% thiourea, were synthesized using a mechanochemical approach[97] in a Retsch MM400 ball mill. Stoichiometric amounts of methylammonium iodide (MAI), lead iodide (PbI$_2$), or methylammonium bromide (MABr) and lead bromide (PbBr$_2$) with and without thiourea were accurately weighed and transferred into a 10 ml milling jar (see below for exact amounts). The jar was loaded with 25 stainless steel balls (5 mm diameter) and 1 ml of cyclohexane, which served as a milling agent. The milling process was conducted at a frequency of 30 Hz for 80 min. After completion of the milling, the cyclohexane was allowed to evaporate by leaving the jar open at ambient conditions for 20 min. The resulting powder was sieved using a 90 μm sieve. Finally, the sieved powders were annealed at 100 °C for 10 min.

MAPbI$_3$ batch 1: 0.239 g of MAI (1.5 mmol) and 0.691 g of PbI$_2$ (1.5 mmol) were milled for 80 min to prepare the black powder. MAPbI$_3$ batch 2: 0.257 g of MAI (1.61 mmol), 0.745 g of PbI$_2$ (1.61 mmol) were milled for 80 min to prepare the black powder. MAPbI$_3$ + thiourea: 0.239 g of MAI (1.5 mmol), 0.691 g of PbI$_2$ (1.5 mmol), and 0.011g of thiourea (0.15 mmol) were milled for 80 min to prepare the black powder.

MAPbBr$_3$ batch 1: 0.212 g of MABr (1.89 mmol) and 0.690 g of PbBr$_2$ (1.89 mmol) were milled for 80 min to prepare the orange powder. MAPbBr$_3$ batch 2: 0.225 g of MABr (2 mmol) and 0.737 g of PbBr$_2$ (2 mmol) were milled for 80 min to prepare the orange powder. MAPbBr$_3$ + thiourea: 0.212 g of MABr (1.89 mmol), 0.690 g of PbBr$_2$ (1.89 mmol), and 0.014 g thiourea (0.18 mmol) were milled for 80 min to prepare the orange powder.

**Mixture of powders for in situ solid-state NMR.** A series of 1:1 molar physical powder mixtures of MAPbI$_3$ and MAPbBr$_3$ was prepared from two separate synthesis batches. For all mixtures, the respective perovskite powders were weighed into a 10 ml stainless steel milling jar.

Batch 1: MAPbI$_3$ / MAPbBr$_3$ (reference); MAPbI$_3$ + 5 mol-% thiourea/MAPbBr$_3$; MAPbI$_3$ + 5 mol-% thiourea / MAPbBr$_3$ + 5 mol-% thiourea

Batch 2: MAPbI$_3$ / MAPbBr$_3$ (reference); MAPbI$_3$ / MAPbBr$_3$/10 mol-% MACl

To prevent any halide exchange during the physical mixing process, the jars were cooled to 77 K using liquid nitrogen. The powders were then physically mixed using a Retsch MM400 ball mill at a frequency of 20 Hz for 20 min (batch 1) or 10 min (batch 2). The jars were re-cooled with liquid nitrogen after the first 10 minutes (batch 1) or 5 min (batch 2) of milling.

## Characterization

**UV photoelectron spectroscopy.** For UPS analysis the samples were transferred from the preparation glove box to the ultra-high vacuum measurement setup under $N_2$ atmosphere, therefore avoiding any air exposure. For the measurement, the monochromatic He lamp (VUV5k, Scienta) was used tuned to the HeIIalpha excitation at 40.8 eV. The emitted photoelectrons were detected with a hemispherical analyzer (Phoibos 100, Specs) using a pass energy of 6 eV. The sample bias was −8 V. Note that the UPS data of the different measurements was aligned such that the valence band onset of all measurements are the same. This was done to eliminate effects of changes in Fermi level position which were present between the different samples.

**J-V characterization of devices.** The solar cell's J-V characteristics were measured outside the glovebox, while being under a constant flow of nitrogen. The measurement was performed using a Keithley 2400 source measurement unit and a 300 W Newport solar simulator (model 91 160, AM1.5G, 100 mW cm$^{-2}$). The simulator was calibrated using a certified IEC 60904-9-compliant Si reference cell from Rera Systems. The measurements were conducted at a scanning speed of 500 mV s$^{-1}$. Stabilized power output was continuously tracked using the same equipment by monitoring the maximum power point (MPP) under AM1.5G illumination.

**EQE measurement of devices.** For EQE measurements, a home-built system was used, where a tunable light source (LOT MSH 150) was employed and current density measurements were taken with a lock-in amplifier (NF Electronic Instruments 5610B). Calibration was executed using a Thorlabs PM100D power meter equipped with an S130VC sensor head.

**Microscopy.** SEM images were obtained by in-lens or secondary electron detector of Gemini 2 SEM from Zeiss at various acceleration speeds. Atomic force microscopy (AFM) topology data was acquired with a Bruker Innova system in tapping mode.

For the powder characterization, synthesized powders were mounted onto standard sample holders using conductive adhesive graphite pads (Plano). To enhance the surface conductivity, the samples were sputter-coated with a 2 nm layer of platinum. Morphological analysis was carried out using a Zeiss Leo 1530 Field Emission Scanning Electron Microscope (FE-SEM), equipped with a Schottky-field-emission cathode, in-lens detector, and SE2 detector. The images were captured at an accelerating voltage of 3.0 kV.

**NMR spectroscopy of inks.** $^{207}Pb$ NMR studies were obtained with a Bruker Avance 400 MHz NMR spectrometer equipped with a BBO-400 MHz, S1, 5 mm probe with z-gradient. Chemical shifts are given with respect to $Pb(CH_3)_4$. For an analyzed range of −500 to 3500 ppm, several spectra with a sweep width of 800 ppm were recorded, which overlapped by at least 100 ppm. For all samples, the probe was locked and shimmed on DMF-d7 using a small, sealed glass tube (2.4/3.0 mm inner/outer diameter) which was placed inside the 5 mm NMR tube.

**Solid-state NMR.** Solid-state NMR spectra were recorded at a magnetic field strength of 14.1 T (600 MHz) on a Bruker Avance HD III spectrometer equipped with a Bruker 3.2 mm HXY MAS NMR probe. Lead nitrate $(Pb(NO_3)_2)$ with a chemical shift of −3494 ppm was used as the reference for $^{207}Pb$ chemical shifts. The samples were spun at a frequency of 5 kHz for magic angle spinning (MAS) using dry air, with the temperature of the air flow varied and calibrated using $Pb(NO_3)_2$. A spin echo sequence (8.33 µs) was employed to suppress ringing, with a recycle delay of 0.5 seconds. Measurements were conducted at a $B_1$ field strength of approximately 150 kHz, with a transmitter offset of 125.66 MHz.

In situ halide mixing experiments were carried out at 80 °C on MAPbI$_3$/MAPbBr$_3$ mixtures, both with and without thiourea or MACl. A series of 1D $^{207}Pb$ NMR spectra were recorded in 5-minute intervals throughout the experiments for 27 h (mixture 1: MAPbI$_3$ + MAPbBr$_3$), 40h (mixture 2: MAPbI$_3$ + MAPbBr$_3$ + 5% thiourea), 3.5 h (mixture 3: MAPbI$_3$ + MAPbBr$_3$ + 10% thiourea), 41 h (mixture 4: MAPbI$_3$ + MAPbBr$_3$) and 41 h (mixture 5: MAPbI$_3$ + MAPbBr$_3$ + 10% MACl)[98].

High-resolution $^1H$ MAS NMR spectra were recorded at a spinning frequency of 62.5 kHz using a commercial Bruker 1.3 mm HXY MAS NMR probe. Adamantane, with a chemical shift of 1.85 ppm, was used as a secondary reference for $^1H$ chemical shifts. A Hahn-echo experiment with a delay of 16 µs and a recycle delay of 70 s was used at a $B_1$ field of 600.15 MHz.

Variable temperature $^1H$ MAS NMR spectra were recorded using a single-pulse sequence at spinning speeds of 20 kHz (Bruker 3.2 mm HXY MAS NMR probe). Measurements were carried out at 30 °C, 50 °C, 60 °C, 70 °C, and 80 °C, with a recycle delay of 70 s. To enhance the relative intensity of the thiourea proton signal compared to the perovskite protons, additional spectra were acquired using a shorter recycle delay of 1 s as the thiourea proton signal possesses as significantly shorter $T_1$ relaxation time. Adamantane (1.85 ppm) was used as a secondary reference for $^1H$ chemical shifts at room temperature. For temperature calibration, the methylammonium proton signals at 3.3 ppm and 6.3 ppm were used to correct for temperature-induced chemical shift changes.

In addition, $^{127}I$ (3/2 → 5/2) Nuclear Quadrupole Resonance (NQR) spectra were acquired using a Bruker Avance HD III spectrometer and a Bruker 3.2 mm HXY MAS probe head, positioned as far away from the magnet as possible. The NQR spectra were recorded at a frequency of 164.09 MHz (3/2 → 5/2 transition equatorial iodine in tetragonal MAPbI$_3$[5]) using a Hahn-echo experiment with as short delays (8.33 µs) as possible, and a recycle delay of 0.1 s ($T_1$ ~ 30 µs).

**FTIR spectroscopy.** 1 M reference inks (PbI$_2$, MAI, PbI$_2$-MAI in DMF:DMSO 4:1 vol) and 1 M thiourea-containing inks (thiourea, PbI$_2$ with 10 mol-% thiourea, MAI with 10 mol-% thiourea, PbI$_2$-MAI with 10 mol-% thiourea in DMF:DMSO 4:1 vol) were spin-coated on silicon wafers (3500 rpm during 35 s) and then annealed at 80 °C during 2–3 min. The FTIR spectra were then recorded on a Bruker Vertex 80 spectrometer.

**Electrical conductance measurement.** The electrical conductance measurement was performed on a custom-built setup utilizing an impedance spectrometer (SI1260) from Solartron, a dosing pump (model: LabF1, head: YZ1515X-PSF) from Shenchen and a generic peristaltic pump. As a mechanical fixture for the ink and the graphite electrodes (SIGRADUR G, glassy carbon), a "cell" was fabricated from PEEK material, separating the electrodes by 3 mm. To avoid cracking of the brittle glassy carbon electrodes, copper plates were introduced to spread the mechanical load of the screws to tighten the cell. The circular contact area of the carbon electrode with the ink is 314.2 mm². Measurements were performed with an alternating voltage of 30 mV$_{peak-peak}$ to avoid electrolysis at a frequency of 31.6 kHz.

**X-ray diffraction.** Thin film XRD was recorded with a Bruker D2 Phaser performing two-theta-theta scans with CuK$_\alpha$ radiation. In situ GIWAXS measurements were performed at DESY (Hamburg, Germany) beamline P08[99] in a dedicated sample environment[64], while thin films were fabricated by the protocols described above. The beam energy was 18 keV, the incidence angle 0. 3° and the beam size was 300 × 100 µm. A Perkin Elmer XRD 1621 detector with a spatial resolution of 200 µm was used to record two-dimensional diffraction patterns with a sample-to-

detector-distance of 700 mm. The integration time during real-time measurements was 100 ms.

Powder X-ray diffraction (PXRD) measurements were performed using CuK$_\alpha$ radiation ($\lambda$ = 1.54 Å) in Bragg-Brentano geometry on a Panalytical EMPYREAN instrument (Malvern Panalytical GmbH, Kassel, Germany). The samples were placed in a spinning sample holder, and data were collected over a $2\theta$ range of 10 – 45° using a PIXcel1D detector.

**Computational settings for DFT calculations.** The initial structure of the tetragonal phase MAPbI$_3$ unit cell was obtained from the hybrid-perovskites collection of the WMD Group[100]. The optimized lattice parameters of the unit cell are as follows: $a$ = 8.695 Å, $b$ = 8.715 Å, and $c$ = 12.834 Å. A slab was constructed from this optimized unit cell, with dimensions of 2 along the $x$, $y$, and $z$ directions. The initial structure of urea was obtained from the Crystallography Open Database (COD, entry number 1008776), while the structure of thiourea was derived by substituting oxygen in urea with sulfur.

Geometric optimizations were performed using the Vienna Ab Initio Simulation Package (VASP)[101], employing the Perdew-Burke-Ernzerhof (PBE) exchange-correlation functional[102] along with the D3 dispersion correction (PBE-D3)[103], using convergence criteria of $10^{-5}$ eV for energy and 0.1 eV Å$^{-1}$ for forces. For the bulk crystal, the k-point grid was 3 with a Gamma mesh. For the slab structure and the single molecule in the vacuum, a 1 k-point grid with a Gamma mesh was used. Electrostatic potential calculations were carried out using the Amsterdam Modeling Suite (release number 2022.103)[104,105], also employing the PBE-D3 functional.

## Data availability
The data generated in this study is provided in the Source Data file. Source data are provided with this paper.

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

## Acknowledgements

We acknowledge the Deutsche Forschungsgemeinschaft (DFG) (within the SPP 2196: grant numbers GR 5505/3-1, RI 1551/15-2, RI 1551/12-2, RI1551/22-1, OL 462/4-2, OL 462/7-1, MA 2359/38-1 and HA 4382/16-1) and the Bundesministerium für Bildung und Forschung (BMBF) (grant number 01DP20008) for financial support. We also thank the BMBF for supporting project 05K19VTA (ERUM-PRO). The research leading to these results has received partial funding from the European Union's Horizon 2020 Program under grant agreement no. 951774 (FOXES). P. W. further thanks the Alexander von Humboldt Foundation for his postdoctoral fellowship. K.B. thanks Prof. Michael Saliba for an inspiring discussion that sparked the initial idea for this project. We acknowledge DESY (Hamburg, Germany), a member of the Helmholtz Association HGF, for the provision of experimental facilities. Parts of this research were carried out at PETRA III, beamline P08. Beamtime was allocated for proposals I-20211642, II-20190761, and I-20221269. This work is partially supported by the grant of the Agency for Innovative Development of the Republic of Uzbekistan, World Bank Project "Modernizing Uzbekistan National Innovation System" (Ref. Nr. REP-24112021/33). H.G. and F.H.B. acknowledge the Northern Bavarian NMR Centre (NBNC) for access to the NMR spectrometer. Z.W. and S.T. acknowledge funding from Vidi (project no. VI.Vidi.213.091) from the Dutch Research Council (NWO). C.M.W. thanks for funding through the Eurostars Eureka project DICE (Project Number 6120). This work was further supported by the European Union's Horizon 2020 research and innovation program under grant agreement No 101075725, TRIUMPH, cofounded by the Swiss State Secretariat for Education, Research and Innovation (SERI).

## Author contributions

K.O.B., L.M., T.M., and T.R. conceived and designed the experiments. K.O.B. and T.R. supervised the project. T.M., G.M., H.W., and S.S. conducted the electric conductivity measurements. T.M. performed the AFM measurements. T.M., F.H.B., C.K., M. Schiffer, G.M., and T.S. carried out the SEM analysis. C.K. and M. Schiffer provided PHP samples. L.M., E.K., V.M., I.Z., F.B., A.H., and F.S. designed and conducted the GIWAXS experiments. T.M., F.Ü., H.W., A.F.K., S.M., and S.F.K. performed the NMR studies. T.M., F.Ü., and E.U. carried out the UV-Vis analysis. C.K. performed the XRD measurements. M.T., P.W., G. Boccarella, D.G., Z.J., and N.A. fabricated solar cells and conducted the J-V characterization. F.H.B. and H.G. designed, conducted, and evaluated the ssNMR analysis. S.D.Ö. and S.O. designed, performed, and evaluated the UPS studies. M.M., O.J.J.R., and J. Harting carried out the phase-field simulations. A.J. and C.M.W. conducted the FTIR analysis. P.W. and G. Boccarella performed the EQE measurements. Z.W., G. Brocks, and S.T. contributed to the DFT simulations. M. Stolterfoht conducted the fast hysteresis measurements. J. He provided chemical insight and sparked the initial investigation. All authors discussed the results and contributed to the writing of the manuscript.

## Funding

## Competing interests
The authors declare no competing interests.

## Additional information

[1]Institute of Electronic Devices, University of Wuppertal, Wuppertal, Germany. [2]Wuppertal Center for Smart Materials & Systems, University of Wuppertal, Wuppertal, Germany. [3]Institute of Applied Physics, University of Tübingen, Tübingen, Germany. [4]Division of Physical Chemistry, Lund University, Lund, Sweden. [5]Institute of Inorganic and Materials Chemistry, University of Cologne, Cologne, Germany. [6]HySPRINT Innovation Lab, Helmholtz-Zentrum Berlin für Materialen und Energie GmbH, Berlin, Germany. [7]Helmholtz Institute Erlangen-Nürnberg for Renewable Energy (HIERN), Forschungszentrum Jülich, Nürnberg, Germany. [8]Inorganic Chemistry III and Bavarian Center for Battery Technology (BayBatt), University of Bayreuth, Bayreuth, Germany. [9]Materials Simulation & Modelling, Department of Applied Physics and Science Education, Eindhoven University of Technology, Eindhoven, The Netherlands. [10]Center for Computational Energy Research, Department of Applied Physics and Science Education, Eindhoven, The Netherlands. [11]Chair for Material and Surface Analysis, University of Wuppertal, Wuppertal, Germany. [12]Eindhoven Institute for Renewable Energy Systems (EIRES) Eindhoven University of Technology, Eindhoven, The Netherlands. [13]Institute of Polymer Chemistry and Physics, Academy of Science of the Republic of Uzbekistan, Tashkent, Uzbekistan. [14]Deutsches Elektronen-Synchrotron DESY, Hamburg, Germany. [15]Institute of Electrical and Microengineering (IEM), Ecole Polytechnique Fédérale de Lausanne (EPFL), Photovoltaics and Thin-Film Electronics Laboratory, Neuchâtel, Switzerland. [16]Department of Science and Technology, Yunnan Agricultural University, Kunming, China. [17]Electronic Engineering Department, The Chinese University of Hong Kong, Hong Kong, SAR, China. [18]Computational Chemical Physics, Faculty of Science and Technology and MESA+ Institute for Nanotechnology, University of Twente, Enschede, The Netherlands. [19]Department of Organic Chemistry, Bergische Universität Wuppertal, Wuppertal, Germany. [20]LISA+, University of Tübingen, Tübingen, Germany. [21]These authors contributed equally: Timo Maschwitz, Lena Merten. ✉e-mail: kotthaus@uni-wuppertal.de; alexander.hinderhofer@uni-tuebingen.de; t.riedl@uni-wuppertal.de; brinkmann@uni-wuppertal.de

