## [Transparent Peer Review file · Nature Communications]

How Crystallization Additives Govern Halide Perovskite Grain Growth.

Corresponding Author: Dr Kai Brinkmann

Version 0:

Reviewer comments:

Reviewer #1

(Remarks to the Author)

This paper by Timo Maschwitz et al. presents a comprehensive investigation into the role of crystallization additives in governing the grain growth of halide perovskites, challenging conventional wisdom and providing new insights into the mechanisms underlying perovskite film formation. The authors employ a multi-faceted methodology, combining precursor chemistry studies, in situ and ex situ characterization techniques. This holistic approach allows them to trace the entire pathway of perovskite formation, from precursor ink to final thin film, providing a framework for understanding the role of additives. Thus, it can be acceptable after addressing the following issues.

1. While the paper focuses on grain growth and crystallization, it is essential to consider the long-term stability of perovskite films. The impact of additives on grain boundary properties and their influence on device stability over extended periods should be explored. Understanding how additives affect defect densities and ionic migration under operational conditions will be crucial for practical applications.
2. The authors propose that additives like thiourea coordinate with lead ions at grain boundaries to enhance ion mobility. However, the exact nature of these coordination interactions and their impact on the electronic structure of the perovskite remain unclear. Further investigation or more detailed spectroscopic analyses could provide deeper insights into these mechanisms..
3. The study focuses on a limited set of additives. Exploring other potential additives or combinations of additives could reveal new pathways for optimizing grain growth and device performance. A broader investigation in this direction could uncover additional strategies for enhancing perovskite film quality and device efficiency.

Reviewer #2

(Remarks to the Author)

his manuscript established a direct link between additive engineering and perovskite post-processing, authors provide evidence that typical crystallization additives do not predominantly impact the nucleation phase, but rather facilitate coarsening grain growth by increasing ion-mobility across grain boundaries. However, there are a number of issues/questions that should be addressed, as listed below.

1. Authors have established a link between additive engineering and perovskite postprocessing, providing a unified framework for advancing material design and process engineering. However, the study only utilized a single additive to validate this conclusion. Given that thiourea's effect on perovskite crystallization has been well documented, it may be insufficient to establish a comprehensive link or provide a unified framework based solely on one additive..
2. In the Results and Discussion section, the statement "we obtained very similar grain sizes independent from the chosen technique and substrate. We consider the results of this set of orienting experiments as a strong indication that the findings presented in this work will have general relevance beyond the choice of substrate, solvent and deposition technique"

appears to be problematic. Perovskite films are polycrystalline materials, and the crystallization process of polycrystalline films exhibits randomness. Factors such as solvent type, spin-coating procedure, annealing temperature, and environmental conditions can significantly influence both the process and outcomes of film crystallization. Therefore, it is unclear how the authors arrived at this conclusion from a variable perspective.

3. The composition of perovskite significantly influences the crystallization process of perovskite films, including grain size, crystallization rate, grain boundary density, and device performance. Additionally, the concentration of perovskite precursor components also plays a crucial role in this process. Authors should thoroughly consider these factors.

4. There are many errors and inappropriate marks in the manuscript and SI, and I would like to ask author to correct them. Figure S11, b, ordinate and abscissa names; some fonts and sizes do not match.

Reviewer #3

(Remarks to the Author)

In this manuscript, the authors demonstrated that crystallization additives primarily influence the coarsening of grain growth by enhancing ion mobility across grain boundaries, rather than significantly affecting the nucleation phase. However, the majority of the presented results are based on the discussion regarding a Lewis-base additive (thiourea). Therefore, the following comments should be considered by the authors during the revision process to further enhance the manuscript's impact:

1. The authors believe they have provided a concept of the mechanism how many popular crystallization additives mediate the grain formation in the final perovskite layer, to strengthen the persuasiveness of this conclusion, at least the most commonly used additive MACl should be discussed. In addition, more mainstream FA-based perovskites rather than MAPbI₃ perovskites should be used as model materials for discussion.

2. In Fig. 3, a measurement setup is employed for in-situ GIWAXS characterization of the perovskite formation process. However, it would be important to illustrate whether the use of infrared heating and direct substrate heating could have any difference on the crystallization process. Clarifying this aspect would strengthen the reliability and interpretation of the experimental results.

3. The authors confirmed that thiourea significantly accelerated the halide migration kinetics across grain boundaries through fitting the solid-state NMR spectra (Fig. 5d), more direct experimental evidence such as conductive AFM testing of ion mobility at grain boundaries can be more convincing, and it is suggested that AFM-IR can be used to confirm the coordination between thiourea to the lead site.

4. In fact, the regulation of alkali metal cations such as Cs⁺ and Rb⁺ on the crystallization/grow process of perovskites has been widely studied and recognized. As mentioned by the author, the crystallization process of MAPbI₃ and FACsPbI₃ is not completely consistent. Thus, the coupling effect between different additives should be discussed and studied, and synergistic introduction rather than single introduction is becoming mainstream.

5. The author provided a lot of evidence in this work, including solid-state and liquid state nuclear magnetic resonance, conductivity measurements, GIWAXS, XRD, and FTIR measurements. However, I believe that only the conclusion that thiourea additives can coarsen MAPbI₃ grains can be drawn, rather than the universal conclusion that additives enable a coarsening-driven grain growth in different perovskite systems. By the way, does the increase in grain size necessarily have a beneficial impact on the performance of photovoltaic devices?

Reviewer #4

(Remarks to the Author)

This manuscript presents a comprehensive study on the role of additives in the perovskite precursor solution and their impact on grain growth in perovskite thin films. The authors provide strong experimental evidence supporting the claim that these additives primarily induce a coarsening growth process rather than retarded nucleation and growth through complex formation with PbI₂. Overall, the manuscript is well-structured, and I would support its publication if the authors could clarify or provide additional evidence on the following key points.

The central conclusion of this study is that grain growth in perovskite thin films occurs via a coarsening process during annealing, facilitated by additives that promote ion migration at grain boundaries. The authors demonstrate that hot pressing induces grain growth by enhancing ion migration at elevated temperatures. Based on this model experiment, they propose that additives remain at the grain boundaries and facilitate ion migration during annealing. While it is reasonable to assume that ion migration increases at high temperatures, the manuscript lacks a clear explanation or direct evidence showing how specific additives, such as thiourea, can enhance ion migration at grain boundaries. Further experimental validation or mechanistic clarification of this process would strengthen the argument.

Version 1:

Reviewer comments:

Reviewer #1

(Remarks to the Author)

As the revisions have satisfactorily addressed my concerns, the current version is acceptable.

Reviewer #2

(Remarks to the Author)

no comments

Reviewer #3

(Remarks to the Author)

The author has solved all the previous questions, it can be published now.

Reviewer #4

(Remarks to the Author)

The authors have adequately addressed the concerns raised by this reviewer. I have no further comments.

Rebuttal

Here is our point-by-point response:

Our answers are *green and italic*

Text in the manuscript is **marked yellow**. Since we sometimes also copied some framing passages for context, the **specific changes** are **highlighted in red**.

Reviewer #1 (Remarks to the Author):

This paper by Timo Maschwitz et al. presents a comprehensive investigation into the role of crystallization additives in governing the grain growth of halide perovskites, challenging conventional wisdom and providing new insights into the mechanisms underlying perovskite film formation. The authors employ a multi-faceted methodology, combining precursor chemistry studies, in situ and ex situ characterization techniques. This holistic approach allows them to trace the entire pathway of perovskite formation, from precursor ink to final thin film, providing a framework for understanding the role of additives.

Answer: We truly appreciate the referee for the positive assessment of our work. We are convinced that addressing the very relevant points raised by the referee substantially improved the quality of our manuscript.

Thus, it can be acceptable after addressing the following issues.

1. While the paper focuses on grain growth and crystallization, it is essential to consider the long-term stability of perovskite films. The impact of additives on grain boundary properties and their influence on device stability over extended periods should be explored. Understanding how additives affect defect densities and ionic migration under operational conditions will be crucial for practical applications.

Reply:

We agree with the referee that an increased long-term stability is a very important property of a well-crystallized perovskite material and should be discussed in a work covering perovskite crystallization. Indeed, in our original manuscript, we already illustrated the inherent instability of our perovskite solar cells, if they are processed without the thiourea additive.

Figure S11: **a, b**, stabilized power conversion efficiencies, **c, d**, EQE before light soaking and **e, f**, the development of the short circuit current density during light soaking, of MAPbI₃ solar cells employing either 0.1 m thiourea (**a, c, e**) or were processed without thiourea additive (**b, d, f**).

To further evaluate the origin of the obvious instability of the devices not employing the thiourea additive, we performed fast hysteresis measurements as we show in the newly added Fig. S12, which reveals significantly increased ionic losses in the devices without thiourea.

Figure S12: Fast hysteresis measurement of MAPbI₃ solar cells both with and w/o thiourea additive showcasing the losses due to mobile ions in both devices. The measurement and analysis have been conducted in analogy to our previous work.⁵⁷ Note that the measurements have been taken after light soaking, because a steady state is required for meaningful results.

We further added to the manuscript:

As solar cells without the additive also showed reduced operation stability, similar as described in earlier work,⁵⁷ we conducted fast hysteresis measurements (Fig. S12) revealing increased ionic losses in devices without thiourea addition, which are likely related to recombination at the higher number of grain boundaries.^{16,58}

Finally, since the main concern of the manuscript is not the device, but the perovskite layer, we evaluated the intrinsic storage stability (in nitrogen) of the perovskite active material. By doing this, we made an interesting observation: while the layers employing thiourea remained largely unchanged, perovskite layers without additive remained only stable on the short term. On the longer run, we observed an increase of the grain size and the formation of needle-like artifacts (Fig. R1).

Figure R1: SEM images of MAPbI₃ layers with and without thiourea stored under inert conditions in the dark.

Indeed, this observation of grain size increase even at room temperature agrees with our hypothesis of grain coarsening driven by ion mobility, since also at room temperature an ion mobility greater than zero has to be assumed and coarsening, even though on much longer timescales than during annealing, is to be expected. While we agree that a detailed analysis of the coarsening at different temperatures and time-scales will be very interesting and could be combined with our findings with planar hot pressing, we feel that such a study is beyond the scope of this work, which rather focuses on grain coarsening on short time-scales. As such we did include the data for the referee in this reply (Fig. R1), but decided to not include it in the manuscript, to keep the focus and take it as subject for a follow-up publication.

2. The authors propose that additives like thiourea coordinate with lead ions at grain boundaries to enhance ion mobility. However, the exact nature of these coordination interactions and their impact on the electronic structure of the perovskite remain unclear. Further investigation or more detailed spectroscopic analyses could provide deeper insights into these mechanisms.

Reply:

The reviewer raises an excellent point, that we used as a motivation to dig deeper into the mechanism of how the additives increase the ion mobility across the grain boundaries. To this end, we followed a three-fold approach: Firstly, we conducted ultraviolet photoelectron spectroscopy (UPS) to complement our FTIR studies and investigated a possible charge transfer involved in the coordination. Secondly, we performed solid state NMR at different temperatures and evaluated the full width half maximum of the ^1H resonance that corresponds to the NH-group of thiourea to obtain insight about the change of mobility of thiourea at elevated temperatures. Finally, we conducted DFT calculations to direct our hypothesis with regard to the possible coordination and ion transport mechanisms. As we consider our new findings very relevant for the paper, we decided to expand our Fig. 5 by including the UPS and solid-state NMR data as Fig. 5e, whilst creating a separate Fig. 6 illustrating the process as we deduct it from those measurements combined with the insights from the DFT calculations. In the main text we added:

To understand the underlying mechanism by which the additive interacts with the perovskite and thereby mediates the increase in ion mobility, we employed additional measurements by ultraviolet photoelectron spectroscopy (UPS, Fig. 5e, top), temperature-dependent ^1H solid-state NMR (Fig. 5e, bottom) and also carried out extended density functional theory (DFT) calculations. Together, these results allow us to propose, for the first time, a comprehensive hypothesis describing how additives promote grain coarsening by interacting with grain boundaries to enhance ion mobility, as illustrated in Fig 6.

To investigate the possibility of charge transfer between the additive and the perovskite at the grain boundary, we performed high energy resolution UPS measurements using a monochromatic He II α excitation at an energy of 40.8 eV. This allows to monitor the binding energy of the Pb 5d $_{5/2}$ semi-core levels close to the surface. The comparison of the Pb 5d $_{5/2}$ peak of a reference sample, as well as films either incorporating thiourea or being post treated by it, are

shown in Fig. 5e (full spectra can be found in Fig. S34). Here, thiourea leads to a shift of the Pb semi-core level toward lower binding energies, indicating electron donation to the lead atoms. This observation aligns with our FTIR data (Fig. S19), allowing us to conclude that thiourea coordinates to lead sites at room temperature. While in a bare perovskite material it is expected that methylammonium (MA) terminates the surface,⁸⁸ the UPS measurement indicated that MA is replaced on the surface by the lead-coordinated thiourea (Fig. 6d), regardless whether the additive was mixed into the precursor or added post-process. Please note that, as we cannot determine the surface coverage of thiourea from our experiments, it is possible that the surface termination is inhomogeneous, comprising a mixture of terminal groups.

To obtain a deeper understanding of the underlying mechanisms that further the ion migration, we then pursued an extensive DFT study (Supplementary Note 5, Table S4, Table S5, Fig. S36 – Fig. S38).

Our DFT calculations reveal that coordination to lead is energetically favourable via the sulphur atom of thiourea, which aligns well with our observations with both FTIR (Fig. S19) and UPS (Fig. S34). Calculations of the adsorption energies comparing additives and solvents (Table S5), also reveal the strong coordination of NMP with the perovskite, which is why a likely scenario is, that thiourea impact occurs, in line with our GIWAXS observations, in the annealing step, where remaining strongly coordinating solvent molecules like NMP are expelled from the perovskite crystal surface and the additives are able to take their place.

At the same time anyhow, at elevated temperatures, solid-state ¹H NMR spectroscopy (Fig. 5c, bottom, Fig. S33) shows a narrowing of the ¹H resonance at around 7.5 ppm corresponding to the NH₂ groups of thiourea (decreased FWHM), indicating greater molecular mobility and suggesting that thiourea also readily detaches from the perovskite surface. This detachment gives rise to two key effects—defect opening and ion shuttling—that both can enhance ion migration, as shown in Fig. 6e. Upon detachment, previously passivated defect sites become exposed. These unpassivated defects or vacancies are known to serve as channels for ion migration, thereby facilitating enhanced ion mobility.^{89,90} Note that in the context of halide mixing, we have earlier observed increased ion mobilities in the presence of ionic liquids.⁸⁶ In the context of ion shuttling, DFT calculations of bonding energies (Table S4 and electrostatic potential distributions (Fig. S37) further reveal that both urea and thiourea exhibit strong coordination between NH₂ groups and iodide, making them potential iodide shuttles that enhance ion transport across grain boundaries. Furthermore, thiourea has a significantly lower detachment energy compared to urea (i.e. a lower adsorption energy, see Table S5), which is consistent with our experimental observation of a more pronounced grain coarsening effect in the presence of thiourea (Fig. 5b). Note that, while we observed a similar overall trend, we found slightly smaller final grain sizes of FA_{0.94}Cs_{0.06}PbI₃ compared to MAPbI₃ upon thiourea addition. This difference could be related to inherently different ion mobilities within the perovskite grains, as also suspected in the context of halide segregation, where FA-based perovskites also show a lower tendency to halide-segregate.⁸⁷

In contrast to urea and thiourea, MACl lacks the ability to function as an ion shuttle. In this case, the primary mechanism enhancing ion migration remains the opening of defects upon MACl

expulsion during annealing.⁹¹ This results in a mobility (μ) sequence of $\mu_{\text{thiourea}} > \mu_{\text{urea}} > \mu_{\text{MACl}}$ (Fig. 6c), which aligns well with the observed grain sizes (Fig. 5b), where a higher ion mobility correlates with higher coarsening rates and concomitantly larger grain sizes. To validate our hypothesis, we extended our solid-state NMR study of halide mixtures (Fig. 5d) by performing an analogous experiment with MACl as the additive as shown in Fig. S33. The results confirmed our hypothesis, showing a five-fold increase in ion mobility for MACl—significant, but still far less than the ~ 50 -fold increase observed for thiourea. We further found a combination of the two additives MACl and thiourea to be substitutive (i.e. the additive that impacts ion mobility the most defines the final appearance of the grains as shown in Fig. S39), which is well in line with our hypothesis, as the amount of possible ion channels is limited by the amount of interface defect states available.

Figure 5: Universal coarsening-mediated growth of perovskite grains. **a**, Evaluated GIWAXS measurements plotting the root-mean-square (rms) of azimuthal intensity fluctuations of characteristic diffraction signals as a function of time, obtained from time-resolved GIWAXS measurements, providing an indirect measure for the grain size. The plots show MAPbI₃ with and without 0.1 M thiourea and 0.1 M urea additive, as well as FA_{0.94}Cs_{0.06}PbI₃ with and without 0.1 M thiourea additive. The insets show SEM images of the respective perovskite layers. **b**, Illustration and results of the post-processing approaches, where the crystallization agent was either spin-coated from isopropanol (IPA) solution on top of the pristine layer with subsequent heating (see also Fig. S23) or planar hot pressing was applied at different temperatures. **c**, Phase-field simulations illustrating a coarsening grain growth with different grain boundary mobilities as a function of time. For details, see Supplementary Note 3. **d**, In situ ion mixing experiment observed by solid-state NMR to track the ion transport kinetics with and without the thiourea additive. (top) Mixing of MAPbI₃ and MAPbBr₃ without additive before and after 210 min annealing at 80 °C illustrating the measurement approach. (bottom) Normalized NMR signal intensities of the [PbI_{6-x}Br_x] environment (denoted as “mixed phase”) during the mixed-halide perovskite phase formation process from physical mixtures of MAPbI₃ and MAPbBr₃ containing 0 %, 5 %, and 10 % thiourea. Respective data was fitted exponentially, in accordance with the Johnson-Mehl-Avrami-Kolmogorov model ($y = 1 - A \cdot \exp(-k x^n)$). In the figure, the parameter k is reported for quantification; the other fitting parameters and further details are given in Table S2 and Supplementary Note 4. A similar study for MACl additive is shown in Fig. S33 e, (Top) Ultraviolet photoelectron spectroscopy (UPS) measurements of MAPbI₃ with and w/o thiourea showing a shift of the Pb 5d_{5/2} semi-core level to higher binding energy in the presence of thiourea, indicating electron transfer from the additive to the lead atoms at the sample surface. The full valence band scans and the corresponding data for MACl treatment are included in Fig. S34. (Bottom) Temperature-dependent solid-state NMR measurements focusing on the ¹H signal of NH₂ groups (~7 ppm).

Figure 6: Atomistic mechanism of how additives can increase ion mobility and grain coarsening. **a, b**, Schematic illustration of the coarsening effect **a**, without and **b**, with the presence of additives at the grain boundary that enhance ion mobility. **c, – e**, Mechanisms of enhanced ion mobility in the presence of additives and elevated temperatures, as derived from FTIR, UPS, solid state NMR measurements and DFT calculations. Both processes, defect opening and ion shuttling, appear to increase ion mobility and further grain coarsening, which is why urea and thiourea, that can act as ion shuttles, generate higher ion mobilities than MAI, which is only expected to open defects. Furthermore, the lower adsorption energy of thiourea in comparison to urea makes it more likely to detach and gain mobility as also indicated in Fig. 5e at elevated temperatures.

In the Introduction we write:

The ion mobility in turn is mediated by the presence of the additive at the grain boundaries of the film. With ultraviolet photoelectron spectroscopy (UPS), density functional theory (DFT) calculation and Fourier transformed infrared spectroscopy (FTIR), we present strong evidence that coordination of the additive to lead sites, along with the opening of interfacial defect states and/or ion shuttling during annealing, are the underlying mechanisms driving additive-mediated grain growth.

In the conclusion:

With a combined effort of DFT calculations, temperature dependent solid-state NMR and UPS, we found strong evidence, that the additives enable a coarsening-driven grain growth, after the nucleation has already taken place, either by opening up defect sites or by additionally acting as an ion-shuttle upon annealing. Our results appear to be universal for various popular crystallization additives with iodide-containing lead-perovskite materials, which includes most of the solar cell materials currently in use.

We further added to the Supporting Information:

Figure S34: UPS data of the reference MAPbI_3 sample, as well as the samples treated with either thiourea or MACl , taken using a monochromatic $\text{He II}\alpha$ excitation. **a**, Shows a wide scan of the VB regions of the reference and thiourea treated samples; the position of the semi-core levels of the $\text{Pb}5d$ doublets are indicated. **b**, focuses on the $\text{Pb } 5d_{5/2}$ peak upon addition and post processing of thiourea. For both the mixed-in and post-treated annealed samples, a clear shift to higher binding energy is evident, indicating an electron donation to the lead orbitals by the additive.

Note that the UPS data of the different measurements was aligned such that the VB onset of all measurements are the same. This was done to eliminate effects of changes in Fermi level position which were present between the different samples. Such a shift in Fermi level also changes the position of the Pb semi-core levels, which would however not be representative for charge transfer between the additive and the Pb atoms. In contrast, the shifts presented here are of the Pb semi-core level with respect to the perovskite VB maximum, and are therefore a clear indication of electron transfer to the surface Pb atoms.

Figure S35: Quantitative ^1H MAS NMR spectra of MAPbI_3 + thiourea at **a**, 62.5 kHz spinning rate, **b**, at 20 kHz spinning rate. **c**, Non-quantitative ^1H MAS NMR spectra of MAPbI_3 + thiourea with reduced recycle delay (1 s) at 20 kHz MAS enhancing the thiourea signal at 7 ppm compared to the perovskite signals. **d**, – **h**, Variable temperature (VT) ^1H MAS NMR spectra (recycle delay: 1 s, 20 kHz MAS) focusing on the signals of the perovskite NH_3^+ group and thiourea are shown at **d**, 30 °C, **e**, 50 °C, **f**, 60 °C, **g**, 70 °C, and **h**, 80 °C. The VT ^1H MAS NMR spectra highlight the temperature-dependent narrowing of the thiourea signal indicating an increase in thiourea mobility at elevated temperatures. The full width half maximum (FWHM) of the thiourea signal is plotted in **i**.

Supplementary Note 5

Bonding energy

To investigate the interaction strength of molecular additives with MA^+ and I^- , the bonding energy is calculated by placing the molecule, with or without the accompanying ion, in a $15 \times 15 \times 15 \text{ \AA}^3$ vacuum box. The bonding energies between the solvent molecules (DMSO and DMF) and MA^+/I^- are also provided as a reference. The results and corresponding structures are presented in Table S4 and Fig. S36.

Table S4: Bonding energy of the solvent and additive molecules with either MA⁺ or I⁻.

molecule	with MA ⁺	with I ⁻
DMF	-1.69	-0.19
DMSO	-1.70	-0.15
urea	-1.61	-0.80
thiourea	-1.45	-0.94

The bonding energies listed in Table S4 suggest that the solvents (DMF, DMSO) as well as the additives (urea, thiourea) tend to have strong bonding interaction with MA⁺. In contrast to the solvent molecules, both urea and thiourea show a clear preference for I⁻. Compared to the solvent molecules, these additives exhibit approximately 0.7 eV lower bonding energy, indicating that, unlike the solvents, they can form bonds with I⁻. This bonding is likely responsible to facilitate the migration of iodide. This supports the hypothesis that thiourea and urea act as ion carriers on the MAPbI₃ surface. Furthermore, the bonding energy of thiourea is lower than that of urea by 0.14 eV, which explains why thiourea performs better as an additive than urea in transporting I⁻ ions during the annealing of the MAPbI₃ films.

Figure S36: Different pairs of DMF, DMSO, urea or thiourea with MA⁺ or I⁻.

Electrostatic potential

The electrostatic potential map (Fig. S37) of each molecule-I⁻ and molecule-MA⁺ pair was calculated to gain further insight into the bonding interactions. The molecule alone was also

computed to investigate its polarity. The results for DMF and DMSO are also included as references.

From the electrostatic potential distribution, it is evident that (thio)urea exhibits stronger polarity compared to DMF and DMSO. In both thiourea and urea, the negative polarity is primarily concentrated around the sulfur and oxygen atoms. As a result, regions further from these atoms exhibit electropositive characteristics. This is characterized by two NH_2 moieties with electropositive characteristic, both interacting with I^- , explaining the strong interaction energies between the two species. In contrast, both solvents exhibit low polarity, showing slight positive polarization of the CH_3 group, and with only one C-H bond interacting with I^- . Therefore, their interaction energy is minimal. Consistent with the bonding energy results, the interaction between (thio)urea and MA^+ is equally strong as those between DMF/DMSO and MA^+ , due to the similar interaction strength of negative O/S atoms with the positive NH_3^+ .

Figure S37: Electrostatic potential distribution of DMF, DMSO, urea, thiourea and their pairs with MA^+ or I^- , color mapping with the potential value, where blue represents the positive potential and red for the negative.

Adsorption Energy

To assess the interaction of urea and thiourea with the MAPbI_3 surface, we calculated their adsorption energies on a tetragonal-phase $2 \times 2 \times 1$ MAPbI_3 slab, while only the additional molecule is allowed to relax during the structural relaxation. The results are presented in Table S5, and the relevant structures are shown in Fig. S38. The adsorption energies of DMF and DMSO are also provided as references.

DMSO exhibits the lowest adsorption energy, followed by urea. The solvent NMP also has a high adsorption energy as -0.73 eV, slightly smaller than urea and DMSO. The adsorption energy of thiourea is much lower than that of urea and DMSO with a value of -0.17 eV. DMF shows a very small adsorption energy, suggesting that it has negligible adsorption on the MAPbI₃ surface. The high adsorption energy of urea indicates that it can remain firmly on the surface, which is beneficial for passivating defects. However, when compared to thiourea, the relatively high adsorption energy also means that urea is less likely to desorb from the surface, limiting its mobility. This reduced mobility further restricts the transfer of I⁻ ions, ultimately diminishing the effectiveness of urea in promoting the formation of larger grains during the annealing process.

Table S5: Adsorption energy of solvent and additives on the MAPbI₃ surface.

molecule	adsorption energy / eV
DMF	-0.08
DMSO	-0.96
urea	-0.88
thiourea	-0.17
NMP	-0.73

The structures shown in Fig. 3 suggest that during the adsorption process, the sulfur or oxygen atoms in thiourea and urea can form a bond with the Pb²⁺ cation on the MAPbI₃ surface, which may explain the electron donation observed in experiments. This interaction between oxygen or sulfur and Pb²⁺ is also present in the solvent.

Figure S38: DMF, DMSO, NMP, urea and thiourea on the MAPbI₃ surface.

In conclusion, the higher polarity of thiourea enables it to form a strong bond with I⁻, while its moderate adsorption energy allows it to desorb easily on the surface during annealing. These two factors make thiourea an ideal ion carrier, enhancing the mobility of I⁻. Additionally, thiourea can improve the stability of MAPbI₃ by passivating the Pb²⁺ cations on the surface. While urea also forms a strong bond with I⁻, its performance as an ion carrier is diminished due to its overly strong bond with Pb²⁺. Therefore, as an additive, thiourea outperforms urea.

We further extended our solid-state NMR experiment by using MACl instead of thiourea, which also showed an increased mobility, though not as high as when thiourea is used, as shown in Fig. S34. We want to note that this trend is in line with the resulting grain sizes we observe, as well as our new findings by UPS, solid state NMR and DFT about the atomistic mechanism of how additives facilitate the increase in ion mobility as we explain in our added Fig. 6.

In the main text we added:

Notably, a similar solid-state NMR halide mixing study for the MACl additive also revealed increased ion mobility in the presence of the additive, albeit far less pronounced (Fig. S33).

In the Supporting Information:

Supplementary Note 4

[...]

Analogously, the evolution of halide ion exchange between powder grains of $\text{MAPbI}_3 + \text{MAPbBr}_3$, $\text{MAPbI}_3 + \text{MAPbBr}_3 + 10\% \text{ MACl}$ (batch 2) was monitored via ^{207}Pb MAS NMR spectroscopy at $80\text{ }^\circ\text{C}$ and evaluated (Fig. S33). The resulting kinetic curves were fitted using the Johnson-Mehl-Avrami-Kolmogorov model,¹¹⁸ with the fitting parameters provided in Table S3. It is important to note that ion diffusion, and consequently the mixing rate, is influenced by both the physical and chemical properties of the powders, such as particle size and defect density, as well as the characteristics of the physical powder mixtures. Since these properties can vary between different synthesis batches, the extracted k values should always be compared relatively to the reference mixture from the same batch.

Figure S33: **a**, ^{207}Pb MAS NMR spectra of a physical mixture of $\text{MAPbI}_3 + \text{MAPbBr}_3$ and $\text{MAPbI}_3 + \text{MAPbBr}_3 + 10\% \text{ MACl}$ before and after annealing at $80\text{ }^\circ\text{C}$ for 500 min. **b**, Corresponding normalized intensity of the formation of $[\text{PbI}_{6-x}\text{Br}_x]$ environments in mixed halide perovskite phases as a function of annealing time at $80\text{ }^\circ\text{C}$. Fits are based on the Johnson-Mehl-Avrami-Kolmogorov model and fit parameters are summarized in Table S3.

Table S3: Fitting parameters for the build-up curve corresponding to the phase formation of $\text{MAPbI}_{1.5}\text{Br}_{1.5}$ and $\text{MAPbI}_{1.5}\text{Br}_{1.5} + 10 \text{ mol-}\% \text{ MACl}$ extracted from the formation of $[\text{PbI}_{6-x}\text{Br}_x]$ environments, as shown in Fig. S33.

starting sample composition	A	k / min^{-1}	n
$\text{MAPbBr}_3 + \text{MAPbI}_3$	1	$4.7 \cdot 10^{-4}$	0.32
$\text{MAPbBr}_3 + \text{MAPbI}_3 + 10 \text{ mol-}\% \text{ MACl}$	1	$1.2 \cdot 10^{-3}$	0.30

- The study focuses on a limited set of additives. Exploring other potential additives or combinations of additives could reveal new pathways for optimizing grain growth and device performance. A broader investigation in this direction could uncover additional strategies for enhancing perovskite film quality and device efficiency.

Reply:

We agree with the referee that extending our study to other additives is a useful test to underline the universality of our hypothesis. In the original manuscript, we already used thiourea (a sulfur

donor), urea (an oxygen donor) and MACI (an organic salt). As we found most lead derivatives to be unsuitable for liquid post-processing, due to limited solubility in orthogonal solvents, we decided to extend our investigation to the popular additive group of pseudo-halide cyanides, that continues to attract serious attention and yields high device efficiencies (Adv. Mater. 2019, 31, 1807029, Adv. Energy Mater. 2021, 11, 2100818, Nature 2024, 628, 306). We further validate our original additives with a $\text{FA}_{0.94}\text{Cs}_{0.06}\text{PbI}_3$ perovskite and finally discuss the effects of simultaneous thiourea and MACI addition. In the manuscript, we discuss the results as follows:

These results motivated us to extend the investigation to the widely used additive MACI, which also has shown promise for post-treatment,^{74,75} as well as another group of additives based on pseudo-halides,^{37,76–78} where we chose ammonium thiocyanate, methylammonium cyanate, and methylammonium thiocyanate for their ability to be dissolved in an orthogonal solvent, as shown in Fig. S24. Notably, previous studies have often attributed grain size enhancement in the context of MACI or pseudo-halides to nucleation effects.⁷⁸ However, our findings demonstrate that similar increases in grain size occur regardless of whether the additives are incorporated during thin film deposition or introduced via post-processing. Validation with $\text{FA}_{0.94}\text{Cs}_{0.06}\text{PbI}_3$ as the perovskite composition while using multiple additives exhibited a similar trend as shown in Fig. S25. Taken together, our findings challenge the applicability of the currently presented nucleation-based explanations, clearly rendering them insufficient to account for the grain growth effects observed with these additives and underscore the universality of the coarsening mechanisms for additive-mediated grain growth that we describe later

Figure S24: Comparison of popular additives used as a mix-in additive (0.1 m) and as a post-processing crystallization agent for MAPbI₃, which clearly shows in the SEM images that all respective crystallization agents increase the grain size regardless of whether they were present during the perovskite nucleation or not.

Figure S25: SEM images showing the grain size development with FA_{0.94}Cs_{0.06}PbI₃ as the perovskite composition, comparing the additive at 0.1 m mixed into a 1 m precursor solution or used post-process, confirming the validity of our findings also when different perovskite cations are used.

With respect to the combination of additives, we observed in our new Fig. S39 that in-

deed the impacts of MACl and thiourea appear to be substitutive, where the effect of thiourea is clearly dominant, while an effect of MACl on the grain size can only be seen without or at very low concentrations of thiourea.

In the manuscript we added:

We further found a combination of the two additives MACl and thiourea to be substitutive (i.e. the additive that impacts ion mobility the most defines the final appearance of the grains as shown in Fig. S39), which is well in line with our hypothesis, as the amount of possible ion channels is limited by the amount of interface defect states available.

Note that, while we observed a similar overall trend, we found slightly smaller final grain sizes of $\text{FA}_{0.94}\text{Cs}_{0.06}\text{PbI}_3$ compared to MAPbI_3 upon thiourea addition. This difference could be related to inherently different ion mobilities within the perovskite grains, as also suspected in the context of halide segregation, where FA-based perovskites also show a lower tendency to halide-segregate.⁸⁷

Figure S39: SEM images of MAPbI₃ layers employing different concentrations of both thiourea and MACI, which clearly shows that the grain morphology is largely dominated by the additive that has the strongest impact on ion mobility.

Reviewer #2:

This manuscript established a direct link between additive engineering and perovskite post-processing, authors provide evidence that typical crystallisation additives do not predominantly impact the nucleation phase, but rather facilitate coarsening grain growth by increasing ion-mobility across grain boundaries. However, there are a number of issues/questions that should be addressed, as listed below.

1. Authors have established a link between additive engineering and perovskite post-processing, providing a unified framework for advancing material design and process engineering. However, the study only utilized a single additive to validate this conclusion. Given that thiourea's effect on perovskite crystallization has been well documented, it may be insufficient to establish a comprehensive link or provide a unified framework based solely on one additive.

Reply: We appreciate that the referee agrees with our conclusion that there is a unified framework for additive engineering that is true regardless if the additive is already present during the nucleation phase (i.e. mixed in the precursor) or is added later by post-processing. We fully agree with the assessment of the referee that only showing this effect for a single additive would not be sufficient to make a claim of universality of our finding. This is why in the original study, we already included the additives urea (an oxygen donor) and methylammonium chloride (organic salt), which showed similar behaviour. We document this in Fig. 5a and b.

Figure 5: Universal coarsening-mediated growth of perovskite grains. **a**, Evaluated GIWAXS measurements plotting the root-mean-square (rms) of azimuthal intensity fluctuations of characteristic diffraction signals as a function of time, obtained from time-resolved GIWAXS measurements, providing an indirect measure for the grain size. The plots show MAPbI_3 with and without 0.1 M thiourea and 0.1 M urea additive, as well as $\text{FA}_{0.94}\text{Cs}_{0.06}\text{PbI}_3$ with and without 0.1 M thiourea additive. The insets show SEM images of the respective perovskite layers. **b**, Illustration and results of the post-processing approaches, where the crystallization agent was either spin-coated from isopropanol (IPA) solution on top of the pristine layer with subsequent heating (see also Fig. S23) or planar hot pressing was applied at different temperatures. **c**, Phase-field simulations illustrating a coarsening grain growth with different grain boundary mobilities as a function of time. For details, see Supplementary Note 3. **d**, In situ ion mixing experiment observed by solid-state NMR to track the ion transport kinetics with and without the thiourea additive. (top) Mixing of MAPbI_3 and MAPbBr_3 without additive before and after 210 min annealing at 80 °C illustrating the measurement approach. (bottom) Normalized NMR signal intensities of the $[\text{PbI}_{6-x}\text{Br}_x]$ environment (denoted as “mixed phase”)

during the mixed-halide perovskite phase formation process from physical mixtures of MAPbI₃ and MAPbBr₃ containing 0 %, 5 %, and 10 % thiourea. Respective data was fitted exponentially, in accordance with the Johnson-Mehl-Avrami-Kolmogorov model ($y = 1 - A \cdot \exp(-k x^n)$). In the figure, the parameter k is reported for quantification; the other fitting parameters and further details are given in Table S2 and Supplementary Note 4. A similar study for MACl additive is shown in Fig. S33 e, (Top) Ultraviolet photoelectron spectroscopy (UPS) measurements of MAPbI₃ with and w/o thiourea showing a shift of the Pb 5d_{5/2} semi-core level to higher binding energy in the presence of thiourea, indicating electron transfer from the additive to the lead atoms at the sample surface. The full valence band scans and the corresponding data for MACl treatment are included in Fig. S34. (Bottom) Temperature-dependent solid-state NMR measurements focusing on the ¹H signal of NH₂ groups (~7 ppm).

Still, we very much agree with the referee that extending our study to other additives is a useful test to underline the universality of our hypothesis, as it has also been brought up by other referees. As we found most lead derivatives to be unsuitable for liquid post processing, due to limited solubility in orthogonal solvents, we decided to extend our investigation to the popular additive group of pseudo-halide cyanides, that continues to attract serious attention and yield high device efficiencies (Adv. Mater. 2019, 31, 1807029, Adv. Energy Mater. 2021, 11, 2100818, Nature 2024, 628, 306). We further validate our original additives with a FA_{0.94}Cs_{0.06}PbI₃ perovskite. In the manuscript we discuss the results as follows:

These results motivated us to extend the investigation to the widely used additive MACl, which also has shown promise for post-treatment,^{74,75} as well as another group of additives based on pseudo-halides,^{37,76–78} where we chose ammonium thiocyanate, methylammonium cyanate, and methylammonium thiocyanate for their ability to be dissolved in an orthogonal solvent, as shown in Fig. S24. Notably, previous studies have often attributed grain size enhancement in the context of MACl or pseudo-halides to nucleation effects.⁷⁸ However, our findings demonstrate that similar increases in grain size occur regardless of whether the additives are incorporated during thin film deposition or introduced via post-processing. Validation with FA_{0.94}Cs_{0.06}PbI₃ as the perovskite composition while using multiple additives exhibited a similar trend as shown in Fig. S25. Taken together, our findings challenge the applicability of the currently presented nucleation-based explanations, clearly rendering them insufficient to account for the grain growth effects observed with these additives and underscore the universality of the coarsening mechanisms for additive-mediated grain growth that we describe later

Figure S24: Comparison of popular additives used as a mix-in additive (0.1 m) and as a post-processing crystallization agent for MAPbI_3 , which clearly shows in the SEM images that all respective crystallization agents increase the grain size regardless of whether they were present during the perovskite nucleation or not.

Figure S25: SEM images showing the grain size development with $\text{FA}_{0.94}\text{Cs}_{0.06}\text{PbI}_3$ as the perovskite composition, comparing the additive at 0.1 m mixed into a 1 m precursor solution or used post-process, confirming the validity of our findings also when different perovskite cations are used.

In the conclusion we added:

Our results appear to be universal for various popular crystallization additives with iodide-containing lead-perovskite materials, which includes most of the solar cell materials currently in use.

We further extended our solid-state NMR experiment by using MACI instead of thiourea, which also showed an increased mobility, though not as high as when thiourea is used, as shown in Fig. S34. We want to note that this trend is in line with the resulting grain sizes we observe, as well as our new findings by UPS, solid state NMR and DFT about the atomistic mechanism of how additives facilitate the increase in ion mobility as we explain in our added Fig. 6.

In the main text we added:

Notably, a similar solid-state NMR halide mixing study for the MACI additive also revealed increased ion mobility in the presence of the additive, albeit far less pronounced (Fig. S33).

In the Supporting Information:

Supplementary Note 4

Analogously, the evolution of halide ion exchange between powder grains of $\text{MAPbI}_3 + \text{MAPbBr}_3$, $\text{MAPbI}_3 + \text{MAPbBr}_3 + 10\% \text{ MACI}$ (batch 2) was monitored via ^{207}Pb MAS NMR spectroscopy at 80°C and evaluated (Fig. S33). The resulting kinetic curves were fitted using the Johnson-Mehl-Avrami-Kolmogorov model,¹¹⁸ with the fitting parameters provided in Table S3. It is important to note that ion diffusion, and consequently the mixing rate, is influenced by both the physical and chemical properties of the powders, such as particle size and defect density, as well as the characteristics of the physical powder mixtures. Since these properties can vary between different synthesis batches, the extracted k values should always be compared relatively to the reference mixture from the same batch.

[...]

Figure S33: **a**, ^{207}Pb MAS NMR spectra of a physical mixture of $\text{MAPbI}_3 + \text{MAPbBr}_3$ and $\text{MAPbI}_3 + \text{MAPbBr}_3 + 10\% \text{ MACI}$ before and after annealing at 80°C for 500 min. **b**, Corresponding normalized intensity of the formation of $[\text{PbI}_{6-x}\text{Br}_x]$ environments in mixed halide perovskite phases as a function of annealing time at 80°C . Fits are based on the Johnson-Mehl-Avrami-Kolmogorov model and fit parameters are summarized in Table S3.

Table S3: Fitting parameters for the build-up curve corresponding to the phase formation of $\text{MAPbI}_{1.5}\text{Br}_{1.5}$ and $\text{MAPbI}_{1.5}\text{Br}_{1.5} + 10 \text{ mol-}\%$ MACl extracted from the formation of $[\text{PbI}_{6-x}\text{Br}_x]$ environments, as shown in Fig. S33.

starting sample composition	A	k / min^{-1}	n
$\text{MAPbBr}_3 + \text{MAPbI}_3$	1	$4.7 \cdot 10^{-4}$	0.32
$\text{MAPbBr}_3 + \text{MAPbI}_3 + 10 \text{ mol-}\%$ MACl	1	$1.2 \cdot 10^{-3}$	0.30

Figure 6: Atomistic mechanism of how additives can increase ion mobility and grain coarsening. a, b, Schematic illustration of the coarsening effect a, without and b, with the presence of additives at the grain boundary that enhance ion mobility. c, – e, Mechanisms of enhanced ion mobility in the presence of additives and elevated temperatures, as derived from FTIR, UPS, solid state NMR measurements and DFT calculations. Both processes, defect opening and ion shuttling, appear to increase ion mobility and further grain coarsening, which is why urea and thiourea, that can act as ion shuttles, generate higher ion mobilities than MACl, which is only expected to open defects. Furthermore, the lower adsorption energy of thiourea in comparison to urea makes it more likely to detach and gain mobility as also indicated in Fig. 5e at elevated temperatures.

In the Results and Discussion section, the statement “we obtained very similar grain sizes independent from the chosen technique and substrate. We consider the results of this set of orienting experiments as a strong indication that the findings presented in this work will have general relevance beyond the choice of substrate, solvent and deposition technique” appears to be problematic. Perovskite films are polycrystalline materials, and the crystallisation process of polycrystalline films exhibits randomness. Factors such as solvent type, spin-coating procedure, annealing temperature, and environmental conditions can significantly influence both the process and outcomes of film crystallization. Therefore, it is unclear how the authors

arrived at this conclusion from a variable perspective.

Reply:

The referee is correct, as we might have phrased this sentence somewhat misleadingly. We didn't mean to imply that there would be no environmental influences on the perovskite layer formation. Indeed, the morphology of the perovskite surface (i.e. roughness, pin-holes, etc.) is very much impacted by the deposition technique, while, at least in our set of experiments, the actual size of the grains appears independent of the deposition technique. Surely also atmospheric conditions and temperature during annealing affect the grain size, as they can have similar effects like additives. Indeed, in Fig. 5b, we directly address the impact of temperature, which naturally enhances the mobility of the ions and enhances coarsening.

We have rephrased the sentence to better specify the claims intended by it:

We consider the results of this set of initial experiments as a strong indication that our findings for the development of the final perovskite grain size will be transferrable to both hydrophilic and hydrophobic substrate types, slow and fast deposition techniques and different popular perovskite solvent systems.

We further extended the validation of this claim later in the manuscript, to also confirm the independence of the deposition technique:

We found the impact of thiourea to be largely independent of the solvent-system (Fig. S8) and the perovskite deposition technique (i.e. drying speed, Fig. S9) and therefore chose the DMF:NMP (7:3 volumetric ratio) solvent system and the gas-quenching protocol as a reference deposition procedure which is known for its low process variation.⁹

Figure S9: Impact of 0.1 M of thiourea additive on MAPbI₃ grain size in dependence of the deposition technique. The SEM data shows the independence of the effect from the solvent drive-out speed (compare Fig. S2).

2. The composition of perovskite significantly influences the crystallization process of perovskite films, including grain size, crystallization rate, grain boundary density, and device performance.

Reply:

We agree with the referee that the perovskite composition is surely highly relevant for the device performance. Having a closer look at the development of the perovskite <110> diffraction signal (MAPbI₃) or the delta phase (which forms during the spin-coating of FA_{0.94}Cs_{0.06}PbI₃ as an intermediate phase) in Fig. 5a, we would also agree that there is some minor impact of the perovskite composition on the initial crystallization / drying process of the perovskite (see Fig. R2).

Figure R2: Development of the <110> reflection of MAPbI₃ and the reflection of the delta phase of FA_{0.94}Cs_{0.06}PbI₃ over time during the gas-quenching procedure resembling the initial crystallization phase, before annealing. For all an exponential and root-function fits starting from the initiation of the gas-quenching procedure are displayed.

Taking the formation dynamic of the initial crystal phase of MAPbI₃ and the delta phase of FA_{0.94}Cs_{0.06}PbI₃ as a proxy for the crystallization dynamics, we plotted them as a function of time and, in an attempt to enable quantification, we used an exponential and a root function to fit the data. Notably for FA_{0.94}Cs_{0.06}PbI₃ an exponential fit, which was our first choice assuming a saturation process, did not provide a good fit, we used the root function as an alternative, which we also use for the coarsening growth. Indeed, we found substantial differences between the perovskite phase of MAPbI₃ and the delta phase of FA_{0.94}Cs_{0.06}PbI₃, where the emergence of the MAPbI₃ phase rather resembles exponential saturation, while the formation of the delta phase rather follows a root-function. Thereby the referee is certainly correct, that the composition and the crystal phase forming during the gas quenching impacts on the initial crystallization. We want to underline that in both cases a clear impact of the additive, which is the key focus of this manuscript, on the crystallization rate is not found. Especially in the presence of thiourea in both MAPbI₃ and FA_{0.94}Cs_{0.06}PbI₃, the apparent substantial change in grain size is not reflected in the observed crystallization dynamics. Indeed, the solvent drive-out speeds (especially anti-solvent, with <1 s, see Fig. S1 and Fig. S2) would be expected to impact the crystallization rate far more severe, while yielding similar results, as shown above.

So, while we cannot exclude some impact of the perovskite composition on the grain size under certain conditions, we conclude, that the impact of the additive after the initial crystallization phase (Fig. 5a), during the annealing step, by far overcompensates compositional effects during the initial crystallization phase. According to our assessment, we can confirm that the effect of additive mediated grain coarsening appears to be universal for different perovskite compositions and largely independent of the exact nature of the initial crystallization phase. On the other hand, we would certainly expect an impact of the composition on the ion mobility within the perovskite grain, that might impact on the outcome of the coarsening, as a secondary effect. While the general mechanism of grain size increase by the additive remains similar, Fig. 5a shows that the final grain size with 0.1 M of thiourea is different for MAPbI₃ and FA_{0.94}Cs_{0.06}PbI₃. To address this, we added to the discussion:

Note that, while we observed a similar overall trend, we found slightly smaller final grain sizes of FA_{0.94}Cs_{0.06}PbI₃ compared to MAPbI₃ upon thiourea addition. This difference could be related to inherently different ion mobilities within the perovskite grains, as also suspected in the context of halide segregation, where FA-based perovskites also show a lower tendency to halide-segregate.⁸⁷

3. Additionally, the concentration of perovskite precursor components also plays a crucial role in this process. Authors should thoroughly consider these factors.

Reply:

We agree with the referee that the concentration of the precursor and hence the overall amount of material deposited during the initial crystallization has an impact on the coarsening of the perovskite grains, which is inherently impacted by both the aspect ratio (i.e. the layer thickness) and the overall supply of material. To validate an increased ion-mobility due to thiourea regardless of layer thickness, we conducted a study comparing the grain sizes of perovskite precursors for varied perovskite precursor concentrations, where we found consistency with the predictions made from coarsening hypothesis.

Figure S20: MAPbI₃ grain sizes with and without 10 mol-% thiourea at different precursor concentrations illustrating the consistency of the grain-size increase of almost an order of magnitude over a wide range of precursor concentrations. Please note that the increasing precursor concentration increases the film thickness and that the coarsening slows down when the average crystal size approaches the film thickness.^{106,107}

In the main text we added:

As a first validation, we investigated the consistency of our findings at different precursor

concentrations, where we found a general dependency of the initial perovskite grain size of the pristine materials on the grain aspect ratio (i.e. grain equivalent radius vs. film thickness, see Fig. S20). This is in line with the inherent dependency of grain coarsening on the aspect ratio of the grains that inherently depends on the layer thickness. Addition of thiourea nevertheless resulted in an increase in grain size of almost an order of magnitude in all cases.

4. There are many errors and inappropriate marks in the manuscript and SI, and I would like to ask author to correct them. Figure S11, b, ordinate and abscissa names; some fonts and sizes do not match.

Reply:

We thank the referee for pointing us to these flaws. We have worked over the manuscript thoroughly aiming to eliminate typos and other mistakes.

Reviewer #3 (Remarks to the Author):

In this manuscript, the authors demonstrated that crystallization additives primarily influence the coarsening of grain growth by enhancing ion mobility across grain boundaries, rather than significantly affecting the nucleation phase. However, the majority of the presented results are based on the discussion regarding a Lewis-base additive (thiourea). Therefore, the following comments should be considered by the authors during the revision process to further enhance the manuscript's impact:

1. The authors believe they have provided a concept of the mechanism how many popular crystallization additives mediate the grain formation in the final perovskite layer, to strengthen the persuasiveness of this conclusion, at least the most commonly used additive MAOI should be discussed. In addition, more mainstream FA-based perovskites rather than MAPbI₃ perovskites should be used as model materials for discussion.

Reply:

The referee is correct that a study merely focussed on MAPbI₃ and only one additive would carry the doubt of general applicability. Therefore, we already included FA_{0.94}Cs_{0.06}PbI₃ as perovskite with an alternative cation, urea (oxygen donor) and MAOI (organic salt) in the original version of the manuscript (Fig. 5a and b).

Figure 5: Universal coarsening-mediated growth of perovskite grains. **a**, Evaluated GIWAXS measurements plotting the root-mean-square (rms) of azimuthal intensity fluctuations of characteristic diffraction signals as a function of time, obtained from time-resolved GIWAXS measurements, providing an indirect measure for the grain size. The plots show MAPbI₃ with and without 0.1 M thiourea and 0.1 M urea additive, as well as FA_{0.94}Cs_{0.06}PbI₃ with and without 0.1 M thiourea additive. The insets show SEM images of the respective perovskite layers. **b**, Illustration and results of the post-processing approaches, where the crystallization agent was either spin-coated from isopropanol (IPA) solution on top of the pristine layer with subsequent heating (see also Fig. S23) or planar hot pressing was applied at different temperatures. **c**, Phase-field simulations illustrating a coarsening grain growth with different grain boundary mobilities as a function of time. For details, see Supplementary Note 3. **d**, In situ ion mixing experiment observed by solid-state NMR to track the ion transport kinetics with and without the thiourea additive. (top) Mixing of MAPbI₃ and MAPbBr₃ without additive before and after 210 min annealing at 80 °C illustrating the measurement approach. (bottom) Normalized NMR signal intensities of the [PbI_{6-x}Br_x] environment (denoted as “mixed phase”)

during the mixed-halide perovskite phase formation process from physical mixtures of MAPbI₃ and MAPbBr₃ containing 0 %, 5 %, and 10 % thiourea. Respective data was fitted exponentially, in accordance with the Johnson-Mehl-Avrami-Kolmogorov model ($y = 1 - A \cdot \exp(-k x^n)$). In the figure, the parameter k is reported for quantification; the other fitting parameters and further details are given in Table S2 and Supplementary Note 4. A similar study for MACl additive is shown in Fig. S33 e, (Top) Ultraviolet photoelectron spectroscopy (UPS) measurements of MAPbI₃ with and w/o thiourea showing a shift of the Pb 5d_{5/2} semi-core level to higher binding energy in the presence of thiourea, indicating electron transfer from the additive to the lead atoms at the sample surface. The full valence band scans and the corresponding data for MACl treatment are included in Fig. S34. (Bottom) Temperature-dependent solid-state NMR measurements focusing on the ¹H signal of NH₂ groups (~7 ppm).

To further address the concern of the referee, we extended our mix-in vs. post-processing study also to FA_{0.94}Cs_{0.06}PbI₃, as well as even more additives (e.g., pseudo-halides) to confirm the universality of our findings. We adapted the discussion in the manuscript as:

These results motivated us to extend the investigation to the widely used additive MACl, which also has shown promise for post-treatment,^{74,75} as well as another group of additives based on pseudo-halides,^{37,76–78} where we chose ammonium thiocyanate, methylammonium cyanate, and methylammonium thiocyanate for their ability to be dissolved in an orthogonal solvent, as shown in Fig. S24. Notably, previous studies have often attributed grain size enhancement in the context of MACl or pseudo-halides to nucleation effects.⁷⁸ However, our findings demonstrate that similar increases in grain size occur regardless of whether the additives are incorporated during thin film deposition or introduced via post-processing. Validation with FA_{0.94}Cs_{0.06}PbI₃ as the perovskite composition while using multiple additives exhibited a similar trend as shown in Fig. S25. Taken together, our findings challenge the applicability of the currently presented nucleation-based explanations, clearly rendering them insufficient to account for the grain growth effects observed with these additives and underscore the universality of the coarsening mechanisms for additive-mediated grain growth that we describe later

Figure S24: Comparison of popular additives used as a mix-in additive (0.1 m) and as a post-processing crystallization agent for MAPbI_3 , which clearly shows in the SEM images that all respective crystallization agents increase the grain size regardless of whether they were present during the perovskite nucleation or not.

Figure S25: SEM images showing the grain size development with $\text{FA}_{0.94}\text{Cs}_{0.06}\text{PbI}_3$ as the perovskite composition, comparing the additive at 0.1 m mixed into a 1 m precursor solution or used post-process, confirming the validity of our findings also when different perovskite cations are used.

2. In Fig. 3, a measurement setup is employed for in-situ GIWAXS characterization of the

perovskite formation process. However, it would be important to illustrate whether the use of infrared heating and direct substrate heating could have any difference on the crystallization process. Clarifying this aspect would strengthen the reliability and interpretation of the experimental results.

Reply:

The referee makes an excellent point. Since it is organizationally not feasible to perform SEM on the exact samples processed at DESY, and compare with hot-plate annealing, we mirrored the setup in our glovebox to provide a comparison between infrared and hot plate annealed samples with and without thiourea. As the overall annealing time was shorter in our GIWAXS experiments, we adapted the annealing time on the hot plate accordingly.

Figure S14: Comparison of MAPbI₃ layers employing 0.1 M of thiourea annealed either by infrared lamp or on the hot plate for 4 minutes. The annealing time on the hot plate was adapted to the experimental conditions during the GIWAXS measurement. While again being slightly different in grain morphology, the grain size and the impact of thiourea on the grain size appears comparable in both scenarios.

We added to the manuscript:

In an effort to access the later stages of the perovskite deposition process (supersaturation, crystallization and annealing), we constructed a setup that enables us to monitor the perovskite formation during spin-coating and different quenching processes by recording in situ GIWAXS data using synchrotron X-ray radiation (Fig. 3a).^{63,64} A comparison of perovskite layers annealed via infrared annealing and on a hot-plate can be found in Fig. S14.

3. The authors confirmed that thiourea significantly accelerated the halide migration kinetics across grain boundaries through fitting the solid-state NMR spectra (Fig. 5d), more direct experimental evidence such as conductive AFM testing of ion mobility at grain boundaries can be more convincing, and it is suggested that AFM-IR can be used to confirm the coordination between thiourea to the lead

Reply:

To better understand the increased ion conductivity across grain boundaries, we decided to use a combination of temperature dependent solid-state NMR, UPS and DFT calculations. By doing so, we were able to shed light on the underlying mechanisms furthering the ion mobility and accordingly substantially extended our discussion in the manuscript.

To understand the underlying mechanism by which the additive interacts with the perovskite and thereby mediates the increase in ion mobility, we employed additional measurements by ultraviolet photoelectron spectroscopy (UPS, Fig. 5e, top), temperature-dependent ^1H solid-state NMR (Fig. 5e, bottom) and also carried out extended density functional theory (DFT) calculations. Together, these results allow us to propose, for the first time, a comprehensive hypothesis describing how additives promote grain coarsening by interacting with grain boundaries to enhance ion mobility, as illustrated in Fig 6.

Note that, while we observed a similar overall trend, we found slightly smaller final grain sizes of $\text{FA}_{0.94}\text{Cs}_{0.06}\text{PbI}_3$ compared to MAPbI_3 upon thiourea addition. This difference could be related to inherently different ion mobilities within the perovskite grains, as also suspected in the context of halide segregation, where FA-based perovskites also show a lower tendency to halide-segregate.⁸⁷

To investigate the possibility of charge transfer between the additive and the perovskite at the grain boundary, we performed high energy resolution UPS measurements using a monochromatic He II α excitation at an energy of 40.8 eV. This allows to monitor the binding energy of the Pb 5d $_{5/2}$ semi-core levels close to the surface. The comparison of the Pb 5d $_{5/2}$ peak of a reference sample, as well as films either incorporating thiourea or being post treated by it, are shown in Fig. 5e (full spectra can be found in Fig. S34). Here, thiourea leads to a shift of the Pb semi-core level toward lower binding energies, indicating electron donation to the lead atoms. This observation aligns with our FTIR data (Fig. S19), allowing us to conclude that thiourea coordinates to lead sites at room temperature. While in a bare perovskite material it is expected that methylammonium (MA) terminates the surface,⁸⁸ the UPS measurement indicated that MA is replaced on the surface by the lead-coordinated thiourea (Fig. 6d), regardless whether the additive was mixed into the precursor or added post-process. Please note that, as we cannot determine the surface coverage of thiourea from our experiments, it is possible that the surface termination is inhomogeneous, comprising a mixture of terminal groups.

To obtain a deeper understanding of the underlying mechanisms that further the ion migration, we then pursued an extensive DFT study (Supplementary Note 5, Table S4, Table S5, Fig. S36 – Fig. S38).

Our DFT calculations reveal that coordination to lead is energetically favourable via the sulphur atom of thiourea, which aligns well with our observations with both FTIR (Fig. S19) and UPS (Fig. S34). Calculations of the adsorption energies comparing additives and solvents (Table S5), also reveal the strong coordination of NMP with the perovskite, which is why a likely scenario is, that thiourea impact occurs, in line with our GIWAXS observations, in the annealing step, where remaining strongly coordinating solvent molecules like NMP are expelled from the perovskite crystal surface and the additives are able to take their place.

At the same time anyhow, at elevated temperatures, solid-state ^1H NMR spectroscopy (Fig. 5c, bottom, Fig. S33) shows a narrowing of the ^1H resonance at around 7.5 ppm corresponding to the NH_2 groups of thiourea (decreased FWHM), indicating greater molecular mobility and suggesting that thiourea also readily detaches from the perovskite surface. This detachment gives rise to two key effects—defect opening and ion shuttling—that both can enhance ion migration, as shown in Fig. 6e. Upon detachment, previously passivated defect sites become exposed. These unpassivated defects or vacancies are known to serve as channels for ion migration, thereby facilitating enhanced ion mobility.^{89,90} Note that in the context of halide mixing, we have earlier observed increased ion mobilities in the presence of ionic liquids.⁸⁶ In the context of ion shuttling, DFT calculations of bonding energies (Table S4 and electrostatic potential distributions (Fig. S37) further reveal that both urea and thiourea exhibit strong coordination between NH_2 groups and iodide, making them potential iodide shuttles that enhance ion transport across grain boundaries. Furthermore, thiourea has a significantly lower detachment energy compared to urea (i.e. a lower adsorption energy, see Table S5), which is consistent with our experimental observation of a more pronounced grain coarsening effect in the presence of thiourea (Fig. 5b). In contrast to urea and thiourea, MACl lacks the ability to function as an ion shuttle. In this case, the primary mechanism enhancing ion migration remains the opening of defects upon MACl expulsion during annealing.⁹¹ This results in a mobility (μ) sequence of $\mu_{\text{thiourea}} > \mu_{\text{urea}} > \mu_{\text{MACl}}$ (Fig. 6c), which aligns well with the observed grain sizes (Fig. 5b), where a higher ion mobility correlates with higher coarsening rates and concomitantly larger grain sizes. To validate our hypothesis, we extended our solid-state NMR study of halide mixtures (Fig. 5d) by performing an analogous experiment with MACl as the additive as shown in Fig. S33. The results confirmed our hypothesis, showing a five-fold increase in ion mobility for MACl—significant, but still far less than the ~ 50 -fold increase observed for thiourea. We further found a combination of the two additives MACl and thiourea to be substitutive (i.e. the additive that impacts ion mobility the most defines the final appearance of the grains as shown in Fig. S39), which is well in line with our hypothesis, as the amount of possible ion channels is limited by the amount of interface defect states available.

In the conclusion we added:

With a combined effort of DFT calculations, temperature dependent solid-state NMR and UPS, we found strong evidence, that the additives enable a coarsening-driven grain growth, after the nucleation has already taken place, either by opening up defect sites or by additionally

acting as an ion-shuttle upon annealing.

Figure 5: Universal coarsening-mediated growth of perovskite grains. a, Evaluated GIWAXS measurements plotting the root-mean-square (rms) of azimuthal intensity fluctuations of characteristic diffraction signals as a function of time, obtained from time-resolved GIWAXS measurements, providing an indirect measure for the grain size. The plots show MAPbI₃ with and without 0.1 M thiourea and 0.1 M urea additive, as well as FA_{0.94}Cs_{0.06}PbI₃ with and without 0.1 M thiourea additive. The insets show SEM images of the respective perovskite layers. **b,** Illustration and results of the post-processing approaches, where the crystallization agent was either spin-coated from isopropanol (IPA) solution on top of the pristine layer with subsequent heating (see also Fig. S23) or planar hot pressing was applied at different temperatures. **c,** Phase-field simulations illustrating a coarsening grain growth with different grain boundary mobilities as a function of time. For details, see Supplementary Note 3. **d,** In situ ion mixing experiment observed by solid-state NMR to track the ion transport kinetics with and without the thiourea additive. (top) Mixing of MAPbI₃ and MAPbBr₃ without additive

before and after 210 min annealing at 80 °C illustrating the measurement approach. (bottom) Normalized NMR signal intensities of the $[\text{PbI}_{6-x}\text{Br}_x]$ environment (denoted as “mixed phase”) during the mixed-halide perovskite phase formation process from physical mixtures of MAPbI_3 and MAPbBr_3 containing 0 %, 5 %, and 10 % thiourea. Respective data was fitted exponentially, in accordance with the Johnson-Mehl-Avrami-Kolmogorov model ($y = 1 - A \cdot \exp(-k x^n)$). In the figure, the parameter k is reported for quantification; the other fitting parameters and further details are given in Table S2 and Supplementary Note 4. A similar study for MACl additive is shown in Fig. S33 e, (Top) Ultraviolet photoelectron spectroscopy (UPS) measurements of MAPbI_3 with and w/o thiourea showing a shift of the $\text{Pb } 5d_{5/2}$ semi-core level to higher binding energy in the presence of thiourea, indicating electron transfer from the additive to the lead atoms at the sample surface. The full valence band scans and the corresponding data for MACl treatment are included in Fig. S34. (Bottom) Temperature-dependent solid-state NMR measurements focusing on the ^1H signal of NH_2 groups (~ 7 ppm).

Figure 6: Atomistic mechanism of how additives can increase ion mobility and grain coarsening. **a, b**, Schematic illustration of the coarsening effect **a**, without and **b**, with the presence of additives at the grain boundary that enhance ion mobility. **c, – e**, Mechanisms of enhanced ion mobility in the presence of additives and elevated temperatures, as derived from FTIR, UPS, solid state NMR measurements and DFT calculations. Both processes, defect opening and ion shuttling, appear to increase ion mobility and further grain coarsening, which is why urea and thiourea, that can act as ion shuttles, generate higher ion mobilities than MACl , which is only expected to open defects. Furthermore, the lower adsorption energy of thiourea in comparison to urea makes it more likely to detach and gain mobility as also indicated in Fig. 5e at elevated temperatures.

We further added to the Supplementary Information:

Figure S34: UPS data of the reference MAPbI₃ sample, as well as the samples treated with either thiourea or MAOI, taken using a monochromatic He II α excitation. **a**, Shows a wide scan of the VB regions of the reference and thiourea treated samples; the position of the semi-core levels of the Pb5d doublets are indicated. **b**, focuses on the Pb 5d_{5/2} peak upon addition and post processing of thiourea. For both the mixed-in and post-treated annealed samples, a clear shift to higher binding energy is evident, indicating an electron donation to the lead orbitals by the additive.

Note that the UPS data of the different measurements was aligned such that the VB onset of all measurements are the same. This was done to eliminate effects of changes in Fermi level position which were present between the different samples. Such a shift in Fermi level also changes the position of the Pb semi-core levels, which would however not be representative for charge transfer between the additive and the Pb atoms. In contrast, the shifts presented here are of the Pb semi-core level with respect to the perovskite VB maximum, and are therefore a clear indication of electron transfer to the surface Pb atoms.

Supplementary Note 4

[...]

Analogously, the evolution of halide ion exchange between powder grains of MAPbI₃ + MAPbBr₃, MAPbI₃ + MAPbBr₃ + 10 % MAOI (batch 2) was monitored via ²⁰⁷Pb MAS NMR spectroscopy at 80 °C and evaluated (Fig. S33). The resulting kinetic curves were fitted using the Johnson-Mehl-Avrami-Kolmogorov model,¹¹⁸ with the fitting parameters provided in Table S3. It is important to note that ion diffusion, and consequently the mixing rate, is

influenced by both the physical and chemical properties of the powders, such as particle size and defect density, as well as the characteristics of the physical powder mixtures. Since these properties can vary between different synthesis batches, the extracted k values should always be compared relatively to the reference mixture from the same batch.

Figure S35: Quantitative ^1H MAS NMR spectra of MAPbI_3 + thiourea at **a**, 62.5 kHz spinning rate, **b**, at 20 kHz spinning rate. **c**, Non-quantitative ^1H MAS NMR spectra of MAPbI_3 + thiourea with reduced recycle delay (1 s) at 20 kHz MAS enhancing the thiourea signal at 7 ppm compared to the perovskite signals. **d**, – **h**, Variable temperature (VT) ^1H MAS NMR spectra (recycle delay: 1 s, 20 kHz MAS) focusing on the signals of the perovskite NH_3^+ group and thiourea are shown at **d**, 30 °C, **e**, 50 °C, **f**, 60 °C, **g**, 70 °C, and **h**, 80 °C. The VT ^1H MAS NMR spectra highlight the temperature-dependent narrowing of the thiourea signal indicating an increase in thiourea mobility at elevated temperatures. The full width half maximum (FWHM) of the thiourea signal is plotted in **i**.

Supplementary Note 5

Bonding energy

To investigate the interaction strength of molecular additives with MA^+ and I^- , the bonding energy is calculated by placing the molecule, with or without the accompanying ion, in a

$15 \times 15 \times 15 \text{ \AA}^3$ vacuum box. The bonding energies between the solvent molecules (DMSO and DMF) and MA^+/I^- are also provided as a reference. The results and corresponding structures are presented in Table S4 and Fig. S36.

Table S4: Bonding energy of the solvent and additive molecules with either MA^+ or I^- .

molecule	with MA^+	with I^-
DMF	-1.69	-0.19
DMSO	-1.70	-0.15
urea	-1.61	-0.80
thiourea	-1.45	-0.94

The bonding energies listed in Table S4 suggest that the solvents (DMF, DMSO) as well as the additives (urea, thiourea) tend to have strong bonding interaction with MA^+ . In contrast to the solvent molecules, both urea and thiourea show a clear preference for I^- . Compared to the solvents, these additives exhibit approximately 0.7 eV lower bonding energy, indicating that, unlike the solvents, they can form bonds with I^- . This bonding is likely responsible to facilitate the migration of iodide. This supports the hypothesis that thiourea and urea act as ion carriers on the MAPbI_3 surface. Furthermore, the bonding energy of thiourea is lower than that of urea by 0.14 eV, which explains why thiourea performs better as an additive than urea in transporting I^- ions during the annealing of the MAPbI_3 films.

Figure S36: Different pairs of DMF, DMSO, urea or thiourea with MA⁺ or I⁻.

Electrostatic potential

The electrostatic potential map (Fig. S37) of each molecule-I⁻ and molecule-MA⁺ pair was calculated to gain further insight into the bonding interactions. The molecule alone was also computed to investigate its polarity. The results for DMF and DMSO are also included as references.

From the electrostatic potential distribution, it is evident that (thio)urea exhibits stronger polarity compared to DMF and DMSO. In both thiourea and urea, the negative polarity is primarily concentrated around the sulfur and oxygen atoms. As a result, regions further from these atoms exhibit electropositive characteristics. This is characterized by two NH₂ moieties with electropositive characteristic, both interacting with I⁻, explaining the strong interaction energies between the two species. In contrast, both solvents exhibit low polarity, showing slight positive polarization of the CH₃ group, and with only one C-H bond interacting with I⁻. Therefore, their interaction energy is minimal. Consistent with the bonding energy results, the interaction between (thio)urea and MA⁺ is equally strong as those between DMF/DMSO and MA⁺, due to the similar interaction strength of negative O/S atoms with the positive NH₃⁺.

Figure S37: Electrostatic potential distribution of DMF, DMSO, urea, thiourea and their pairs with MA^+ or I^- , color mapping with the potential value, where blue represents the positive potential and red for the negative.

Adsorption Energy

To assess the interaction of urea and thiourea with the MAPbI_3 surface, we calculated their adsorption energies on a tetragonal-phase $2 \times 2 \times 1$ MAPbI_3 slab, while only the additional molecule is allowed to relax during the structural relaxation. The results are presented in Table S5, and the relevant structures are shown in Fig. S38. The adsorption energies of DMF and DMSO are also provided as references.

DMSO exhibits the lowest adsorption energy, followed by urea. The solvent NMP also has a high adsorption energy as -0.73 eV, slightly smaller than urea and DMSO. The adsorption energy of thiourea is much lower than that of urea and DMSO with a value of -0.17 eV. DMF shows a very small adsorption energy, suggesting that it has negligible adsorption on the MAPbI_3 surface. The high adsorption energy of urea indicates that it can remain firmly on the surface, which is beneficial for passivating defects. However, when compared to thiourea, the relatively high adsorption energy also means that urea is less likely to desorb from the surface, limiting its mobility. This reduced mobility further restricts the transfer of I^- ions, ultimately diminishing the effectiveness of urea in promoting the formation of larger grains during the annealing process.

Table S5: Adsorption energy of solvent and additives on the MAPbI_3 surface.

molecule	adsorption energy / eV
DMF	-0.08
DMSO	-0.96
urea	-0.88
thiourea	-0.17
NMP	-0.73

The structures shown in Fig. 3 suggest that during the adsorption process, the sulfur or oxygen atoms in thiourea and urea can form a bond with the Pb^{2+} cation on the MAPbI_3 surface, which may explain the electron donation observed in experiments. This interaction between oxygen or sulfur and Pb^{2+} is also present in the solvent.

Figure S38: DMF, DMSO, NMP, urea and thiourea on the MAPbI_3 surface.

In conclusion, the higher polarity of thiourea enables it to form a strong bond with I^- , while its moderate adsorption energy allows it to desorb easily on the surface during annealing. These two factors make thiourea an ideal ion carrier, enhancing the mobility of I^- . Additionally, thiourea can improve the stability of MAPbI_3 by passivating the Pb^{2+} cations on the surface. While urea also forms a strong bond with I^- , its performance as an ion carrier is diminished due to its overly strong bond with Pb^{2+} . Therefore, as an additive, thiourea outperforms urea.

Figure S33: **a**, ^{207}Pb MAS NMR spectra of a physical mixture of $\text{MAPbI}_3 + \text{MAPbBr}_3$ and $\text{MAPbI}_3 + \text{MAPbBr}_3 + 10\% \text{MACl}$ before and after annealing at $80\text{ }^\circ\text{C}$ for 500 min. **b**, Corresponding normalized intensity of the formation of $[\text{PbI}_{6-x}\text{Br}_x]$ environments in mixed halide perovskite phases as a function of annealing time at $80\text{ }^\circ\text{C}$. Fits are based on the Johnson-Mehl-Avrami-Kolmogorov model and fit parameters are summarized in Table S3.

Table S3: Fitting parameters for the build-up curve corresponding to the phase formation of $\text{MAPbI}_{1.5}\text{Br}_{1.5}$ and $\text{MAPbI}_{1.5}\text{Br}_{1.5} + 10\text{ mol-}\% \text{MACl}$ extracted from the formation of $[\text{PbI}_{6-x}\text{Br}_x]$ environments, as shown in Fig. S33.

starting sample composition	A	k / min^{-1}	n
$\text{MAPbBr}_3 + \text{MAPbI}_3$	1	$4.7 \cdot 10^{-4}$	0.32
$\text{MAPbBr}_3 + \text{MAPbI}_3 + 10\text{ mol-}\% \text{MACl}$	1	$1.2 \cdot 10^{-3}$	0.30

- In fact, the regulation of alkali metal cations such as Cs^+ and Rb^+ on the crystallization/grow process of perovskites has been widely studied and recognized. As mentioned by the author, the crystallization process of MAPbI_3 and FACsPbI_3 is not completely consistent. Thus, the coupling effect between different additives should be discussed and studied, and synergistic introduction rather than single introduction is becoming mainstream.

Reply:

While we see the point of the referee, we cannot confirm the impression that cesium would act as a crystallization agent similar to thiourea or MACl. Indeed, we find a striking similarity between both $\text{FA}_{0.94}\text{Cs}_{0.06}\text{PbI}_3$ and MAPbI_3 in terms of grain size and impact of additives as we have put together in a separate Fig. R3.

Figure R3: Comparison of SEM images of MAPbI₃ and FA_{0.94}Cs_{0.06}PbI₃ thin films both with and without 0.1 M thiourea additive.

Nevertheless, we think the point the referee makes is very interesting, especially in the light of our new finding about the atomistic origin of the additive influence. Given that any interface defect or vacancy can only be occupied once and assuming a sufficient supply of additive molecules, one would rather expect a substitutional than an additive effect of crystallization agents. To this end, we chose MACl and thiourea for our study, as they represent the strongest and the weakest increase in ion mobility covered in this work.

As we observed in our new Fig. S39, indeed the impacts of MACl and thiourea appear to be substitutive, where the effect of thiourea is clearly dominant, while an effect of MACl on the grain size can only be seen without or at very low concentrations of thiourea.

In the manuscript we added:

We further found a combination of the two additives MACl and thiourea to be substitutive (i.e. the additive that impacts ion mobility the most defines the final appearance of the grains as shown in Fig. S39), which is well in line with our hypothesis, as the amount of possible ion channels is limited by the amount of interface defect states available.

Figure S39: SEM images of MAPbI₃ layers employing different concentrations of both thiourea and MACI, which clearly shows that the grain morphology is largely dominated by the additive that has the strongest impact on ion mobility.

- The author provided a lot of evidence in this work, including solid-state and liquid state nuclear magnetic resonance, conductivity measurements, GIWAXS, XRD, and FTIR measurements. However, I believe that only the conclusion that thiourea additives can coarsen MAPbI₃ grains can be drawn, rather than the universal conclusion that additives enable a coarsening-driven grain growth in different perovskite systems.

Reply:

We agree that our original manuscript was lacking some essential proofs and insights to support the general validity of our conclusions. In light of the added experimental data, that we have presented as response to the referee's valid points above, we believe that our results are

of general relevance.

6. By the way, does the increase in grain size necessarily have a beneficial impact on the performance of photovoltaic devices?

Reply:

Please note that we never claimed that an increase in grain size is always beneficial for the device performance. However, the grain boundaries are well known hot spots for non-radiative recombination originating from the inherent interruption of the crystal lattice and concomitant defects and vacancies. While eliminating the grain boundary in the first place is probably the most straightforward strategy to overcome losses induced by the grain boundaries, it is certainly not the only one. While our record devices (e.g. Nature 604, 280–286 (2022)) and plenty of other examples in literature (e.g. Nature 628, 299–305 (2024)) as well the devices that we have shown in this article definitely profit from the grain-boundary reduction, defect states can certainly also be deactivated by other means as e.g. shown in Sci. Adv.8, eabq4524 (2022). To clarify this, we added:

Larger perovskite grains reduce the density of grain boundaries in the final perovskite layer leading to a reduction of deep trap states and non-radiative recombination—key factors influencing solar cell device performance.^{14–16} While there are also other means to mitigate the impact of grain boundaries,¹⁷ reduction of the number of grain boundaries by increasing the grain size is the most straightforward and highly popular approach.

Indeed, we do believe that a too large grain size, also can have its downsides – for example, even though showing record efficiency, our recently published perovskite-organic tandem relied on over 100 nm of PCBM to cover the roughness of the perovskite layer and avoid detrimental effects by a direct contact between perovskite and metal oxide (see Figure S41 in Nature 628, 299–305 (2024)).

In light of this area of conflict, we feel it is even more crucial to understand the mechanism that causes the grains to grow and govern the final grain size. After all, the aim should be to control the perovskite grain formation for tailored device design and not being forced to tailor the devices adapting to the way how a certain perovskite forms. We believe our manuscript presents a substantial step towards actually achieving control and moving beyond heuristics, which might be essential for the success of the perovskite technology.

Reviewer #4 (Remarks to the Author):

This manuscript presents a comprehensive study on the role of additives in the perovskite precursor solution and their impact on grain growth in perovskite thin films. The authors provide strong experimental evidence supporting the claim that these additives primarily induce a coarsening growth process rather than retarded nucleation and growth through complex formation with PbI_2 . Overall, the manuscript is well-structured, and I would support its publication if the authors could clarify or provide additional evidence on the following key points.

Reply:

We appreciate the referee's positive assessment of our work, and we hope that our additional experimental and theoretical data will be useful to address the concerns of the referee.

The central conclusion of this study is that grain growth in perovskite thin films occurs via a coarsening process during annealing, facilitated by additives that promote ion migration at grain boundaries. The authors demonstrate that hot pressing induces grain growth by enhancing ion migration at elevated temperatures. Based on this model experiment, they propose that additives remain at the grain boundaries and facilitate ion migration during annealing. While it is reasonable to assume that ion migration increases at high temperatures, the manuscript lacks a clear explanation or direct evidence showing how specific additives, such as thiourea, can enhance ion migration at grain boundaries. Further experimental validation or mechanistic clarification of this process would strengthen the argument.

Reply:

We agree with the referee's general concern that the data presented and the depth of understanding of the mechanism in the original manuscript was not yet enough. During the review process, we therefore performed a plethora of additional experiments and calculations aiming to deepen our understanding and validate the transferability of our findings.

We unravelled the underlying mechanism of the increased ion mobility by a threefold approach: temperature-dependent solid state NMR, UPS and DFT calculations and added substantial discussion to the manuscript:

To understand the underlying mechanism by which the additive interacts with the perovskite and thereby mediates the increase in ion mobility, we employed additional measurements by ultraviolet photoelectron spectroscopy (UPS, Fig. 5e, top), temperature-dependent ^1H solid-state NMR (Fig. 5e, bottom) and also carried out extended density functional theory (DFT) calculations. Together, these results allow us to propose, for the first time, a comprehensive hypothesis describing how additives promote grain coarsening by interacting with grain boundaries to enhance ion mobility, as illustrated in Fig 6.

To investigate the possibility of charge transfer between the additive and the perovskite at the grain boundary, we performed high energy resolution UPS measurements using a monochromatic He II α excitation at an energy of 40.8 eV. This allows to monitor the binding energy of the Pb 5d_{5/2} semi-core levels close to the surface. The comparison of the Pb 5d_{5/2} peak of a reference sample, as well as films either incorporating thiourea or being post treated by it, are shown in Fig. 5e (full spectra can be found in Fig. S34). Here, thiourea leads to a shift of the Pb semi-core level toward lower binding energies, indicating electron donation to the lead atoms. This observation aligns with our FTIR data (Fig. S19), allowing us to conclude that thiourea coordinates to lead sites at room temperature. While in a bare perovskite material it is expected that methylammonium (MA) terminates the surface,⁸⁸ the UPS measurement indicated that MA is replaced on the surface by the lead-coordinated thiourea (Fig. 6d), regardless whether the additive was mixed into the precursor or added post-process. Please note that, as we cannot determine the surface coverage of thiourea from our experiments, it is possible that the surface termination is inhomogeneous, comprising a mixture of terminal groups.

To obtain a deeper understanding of the underlying mechanisms that further the ion migration, we then pursued an extensive DFT study (Supplementary Note 5, Table S4, Table S5, Fig. S36 – Fig. S38).

Our DFT calculations reveal that coordination to lead is energetically favourable via the sulphur atom of thiourea, which aligns well with our observations with both FTIR (Fig. S19) and UPS (Fig. S34). Calculations of the adsorption energies comparing additives and solvents (Table S5), also reveal the strong coordination of NMP with the perovskite, which is why a likely scenario is, that thiourea impact occurs, in line with our GIWAXS observations, in the annealing step, where remaining strongly coordinating solvent molecules like NMP are expelled from the perovskite crystal surface and the additives are able to take their place.

At the same time anyhow, at elevated temperatures, solid-state ¹H NMR spectroscopy (Fig. 5c, bottom, Fig. S33) shows a narrowing of the ¹H resonance at around 7.5 ppm corresponding to the NH₂ groups of thiourea (decreased FWHM), indicating greater molecular mobility and suggesting that thiourea also readily detaches from the perovskite surface. This detachment gives rise to two key effects—defect opening and ion shuttling—that both can enhance ion migration, as shown in Fig. 6e. Upon detachment, previously passivated defect sites become exposed. These unpassivated defects or vacancies are known to serve as channels for ion migration, thereby facilitating enhanced ion mobility.^{89,90} Note that in the context of halide mixing, we have earlier observed increased ion mobilities in the presence of ionic liquids.⁸⁶ In the context of ion shuttling, DFT calculations of bonding energies (Table S4 and electrostatic potential distributions (Fig. S37) further reveal that both urea and thiourea exhibit strong coordination between NH₂ groups and iodide, making them potential iodide shuttles that enhance ion transport across grain boundaries. Furthermore, thiourea has a significantly lower detachment energy compared to urea (i.e. a lower adsorption energy, see Table S5), which is consistent with our experimental observation of a more pronounced grain coarsening effect in the presence of thiourea (Fig. 5b). Note that, while we observed a similar overall trend, we found slightly smaller final grain sizes of FA_{0.94}Cs_{0.06}PbI₃ compared to MAPbI₃ upon

thiourea addition. This difference could be related to inherently different ion mobilities within the perovskite grains, as also suspected in the context of halide segregation, where FA-based perovskites also show a lower tendency to halide-segregate.⁸⁷

In contrast to urea and thiourea, MACl lacks the ability to function as an ion shuttle. In this case, the primary mechanism enhancing ion migration remains the opening of defects upon MACl expulsion during annealing.⁹¹ This results in a mobility (μ) sequence of $\mu_{\text{thiourea}} > \mu_{\text{urea}} > \mu_{\text{MACl}}$ (Fig. 6c), which aligns well with the observed grain sizes (Fig. 5b), where a higher ion mobility correlates with higher coarsening rates and concomitantly larger grain sizes. To validate our hypothesis, we extended our solid-state NMR study of halide mixtures (Fig. 5d) by performing an analogous experiment with MACl as the additive as shown in Fig. S33. The results confirmed our hypothesis, showing a five-fold increase in ion mobility for MACl—significant, but still far less than the ~ 50 -fold increase observed for thiourea.

In the conclusion we added:

With a combined effort of DFT calculations, temperature dependent solid-state NMR and UPS, we found strong evidence, that the additives enable a coarsening-driven grain growth, after the nucleation has already taken place, either by opening up defect sites or by additionally acting as an ion-shuttle upon annealing.

Figure 5: Universal coarsening-mediated growth of perovskite grains. **a**, Evaluated GIWAXS measurements plotting the root-mean-square (rms) of azimuthal intensity fluctuations of characteristic diffraction signals as a function of time, obtained from time-resolved GIWAXS measurements, providing an indirect measure for the grain size. The plots show MAPbI_3 with and without 0.1 M thiourea and 0.1 M urea additive, as well as $\text{FA}_{0.94}\text{Cs}_{0.06}\text{PbI}_3$ with and without 0.1 M thiourea additive. The insets show SEM images of the respective perovskite layers. **b**, Illustration and results of the post-processing approaches, where the crystallization agent was either spin-coated from isopropanol (IPA) solution on top of the pristine layer with subsequent heating (see also Fig. S23) or planar hot pressing was applied at different temperatures. **c**, Phase-field simulations illustrating a coarsening grain growth with different grain boundary mobilities as a function of time. For details, see Supplementary Note 3. **d**, In situ ion mixing experiment observed by solid-state NMR to track the ion transport kinetics with and without the thiourea additive. (top) Mixing of MAPbI_3 and MAPbBr_3 without additive before and after 210 min annealing at 80 °C illustrating the measurement approach. (bottom) Normalized NMR signal intensities of the $[\text{Pb}_{6-x}\text{Br}_x]$ environment (denoted as “mixed phase”)

during the mixed-halide perovskite phase formation process from physical mixtures of MAPbI₃ and MAPbBr₃ containing 0 %, 5 %, and 10 % thiourea. Respective data was fitted exponentially, in accordance with the Johnson-Mehl-Avrami-Kolmogorov model ($y = 1 - A \cdot \exp(-k x^n)$). In the figure, the parameter k is reported for quantification; the other fitting parameters and further details are given in Table S2 and Supplementary Note 4. A similar study for MAI additive is shown in Fig. S33 e, (Top) Ultraviolet photoelectron spectroscopy (UPS) measurements of MAPbI₃ with and w/o thiourea showing a shift of the Pb 5d_{5/2} semi-core level to higher binding energy in the presence of thiourea, indicating electron transfer from the additive to the lead atoms at the sample surface. The full valence band scans and the corresponding data for MAI treatment are included in Fig. S34. (Bottom) Temperature-dependent solid-state NMR measurements focusing on the ¹H signal of NH₂ groups (~7 ppm).

Figure 6: Atomistic mechanism of how additives can increase ion mobility and grain coarsening. **a, b**, Schematic illustration of the coarsening effect **a**, without and **b**, with the presence of additives at the grain boundary that enhance ion mobility. **c, – e**, Mechanisms of enhanced ion mobility in the presence of additives and elevated temperatures, as derived from FTIR, UPS, solid state NMR measurements and DFT calculations. Both processes, defect opening and ion shuttling, appear to increase ion mobility and further grain coarsening, which is why urea and thiourea, that can act as ion shuttles, generate higher ion mobilities than MAI, which is only expected to open defects. Furthermore, the lower adsorption energy of thiourea in comparison to urea makes it more likely to detach and gain mobility as also indicated in Fig. 5e at elevated temperatures.

We further added to the Supplementary Information:

Figure S34: UPS data of the reference MAPbI₃ sample, as well as the samples treated with either thiourea or MACI, taken using a monochromatic He II α excitation. **a**, Shows a wide scan of the VB regions of the reference and thiourea treated samples; the position of the semi-core levels of the Pb5d doublets are indicated. **b**, focuses on the Pb 5d_{5/2} peak upon addition and post processing of thiourea. For both the mixed-in and post-treated annealed samples, a clear shift to higher binding energy is evident, indicating an electron donation to the lead orbitals by the additive.

Note that the UPS data of the different measurements was aligned such that the VB onset of all measurements are the same. This was done to eliminate effects of changes in Fermi level position which were present between the different samples. Such a shift in Fermi level also changes the position of the Pb semi-core levels, which would however not be representative for charge transfer between the additive and the Pb atoms. In contrast, the shifts presented here are of the Pb semi-core level with respect to the perovskite VB maximum, and are therefore a clear indication of electron transfer to the surface Pb atoms.

Figure S35: Quantitative ^1H MAS NMR spectra of MAPbI_3 + thiourea at **a**, 62.5 kHz spinning rate, **b**, at 20 kHz spinning rate. **c**, Non-quantitative ^1H MAS NMR spectra of MAPbI_3 + thiourea with reduced recycle delay (1 s) at 20 kHz MAS enhancing the thiourea signal at 7 ppm compared to the perovskite signals. **d**, – **h**, Variable temperature (VT) ^1H MAS NMR spectra (recycle delay: 1 s, 20 kHz MAS) focusing on the signals of the perovskite NH_3^+ group and thiourea are shown at **d**, 30 °C, **e**, 50 °C, **f**, 60 °C, **g**, 70 °C, and **h**, 80 °C. The VT ^1H MAS NMR spectra highlight the temperature-dependent narrowing of the thiourea signal indicating an increase in thiourea mobility at elevated temperatures. The full width half maximum (FWHM) of the thiourea signal is plotted in **i**.

Experimental Details

[...]

Computational settings for DFT calculations

The initial structure of the tetragonal phase MAPbI_3 unit cell was obtained from the hybrid-perovskites collection of the WMD Group.¹⁰⁰ The optimized lattice parameters of the unit cell are as follows: $a = 8.695 \text{ \AA}$, $b = 8.715 \text{ \AA}$, and $c = 12.834 \text{ \AA}$. A slab was constructed from this optimized unit cell, with dimensions of 2 along the x , y , and z directions. The initial structure of urea was obtained from the Crystallography Open Database (COD, entry number 1008776),

while the structure of thiourea was derived by substituting oxygen in urea with sulfur. Geometric optimizations were performed using the Vienna Ab Initio Simulation Package (VASP),¹⁰¹ employing the Perdew-Burke-Ernzerhof (PBE) exchange-correlation functional¹⁰² along with the D3 dispersion correction (PBE-D3),¹⁰³ using convergence criteria of 10^{-5} eV for energy and $0.1 \text{ eV } \text{\AA}^{-1}$ for forces. For the bulk crystal, the k-point grid was 3 with a Gamma mesh. For the slab structure and the single molecule in the vacuum, a 1 k-point grid with a Gamma mesh was used. Electrostatic potential calculations were carried out using the Amsterdam Modeling Suite (release number 2022.103),^{104,105} also employing the PBE-D3 functional.

[...]

Supplementary Note 5

Bonding energy

To investigate the interaction strength of molecular additives with MA^+ and I^- , the bonding energy is calculated by placing the molecule, with or without the accompanying ion, in a $15 \times 15 \times 15 \text{ \AA}^3$ vacuum box. The bonding energies between the solvent molecules (DMSO and DMF) and MA^+/I^- are also provided as a reference. The results and corresponding structures are presented in Table S4 and Fig. S36.

Table S4: Bonding energy of the solvent and additive molecules with either MA^+ or I^- .

molecule	with MA^+	with I^-
DMF	-1.69	-0.19
DMSO	-1.70	-0.15
urea	-1.61	-0.80
thiourea	-1.45	-0.94

The bonding energies listed in Table S4 suggest that the solvents (DMF, DMSO) as well as the additives (urea, thiourea) tend to have strong bonding interaction with MA^+ . In contrast to the solvent molecules, both urea and thiourea show a clear preference for I^- . Compared to the solvents, these additives exhibit approximately 0.7 eV lower bonding energy, indicating that, unlike the solvents, they can form bonds with I^- . This bonding is likely responsible to facilitate the migration of iodide. This supports the hypothesis that thiourea and urea act as ion carriers on the MAPbI_3 surface. Furthermore, the bonding energy of thiourea is lower than that of urea by 0.14 eV, which explains why thiourea performs better as an additive than urea in transporting I^- ions during the annealing of the MAPbI_3 films.

Figure S36: Different pairs of DMF, DMSO, urea or thiourea with MA⁺ or I⁻.

Electrostatic potential

The electrostatic potential map (Fig. S37) of each molecule-I⁻ and molecule-MA⁺ pair was calculated to gain further insight into the bonding interactions. The molecule alone was also computed to investigate its polarity. The results for DMF and DMSO are also included as references.

From the electrostatic potential distribution, it is evident that (thio)urea exhibits stronger polarity compared to DMF and DMSO. In both thiourea and urea, the negative polarity is primarily concentrated around the sulfur and oxygen atoms. As a result, regions further from these atoms exhibit electropositive characteristics. This is characterized by two NH₂ moieties with electropositive characteristic, both interacting with I⁻, explaining the strong interaction energies between the two species. In contrast, both solvents exhibit low polarity, showing slight positive polarization of the CH₃ group, and with only one C-H bond interacting with I⁻. Therefore, their interaction energy is minimal. Consistent with the bonding energy results, the interaction between (thio)urea and MA⁺ is equally strong as those between DMF/DMSO and MA⁺, due to the similar interaction strength of negative O/S atoms with the positive NH₃⁺.

Figure S37: Electrostatic potential distribution of DMF, DMSO, urea, thiourea and their pairs with MA⁺ or I⁻, color mapping with the potential value, where blue represents the positive potential and red for the negative.

Adsorption Energy

To assess the interaction of urea and thiourea with the MAPbI₃ surface, we calculated their adsorption energies on a tetragonal-phase 2×2×1 MAPbI₃ slab, while only the additional molecule is allowed to relax during the structural relaxation. The results are presented in Table S5, and the relevant structures are shown in Fig. S38. The adsorption energies of DMF and DMSO are also provided as references.

DMSO exhibits the lowest adsorption energy, followed by urea. The solvent NMP also has a high adsorption energy as -0.73 eV, slightly smaller than urea and DMSO. The adsorption energy of thiourea is much lower than that of urea and DMSO with a value of -0.17 eV. DMF shows a very small adsorption energy, suggesting that it has negligible adsorption on the MAPbI₃ surface. The high adsorption energy of urea indicates that it can remain firmly on the surface, which is beneficial for passivating defects. However, when compared to thiourea, the relatively high adsorption energy also means that urea is less likely to desorb from the surface, limiting its mobility. This reduced mobility further restricts the transfer of I⁻ ions, ultimately diminishing the effectiveness of urea in promoting the formation of larger grains during the annealing process.

Table S5: Adsorption energy of solvent and additives on the MAPbI₃ surface.

molecule	adsorption energy / eV
DMF	-0.08
DMSO	-0.96
urea	-0.88
thiourea	-0.17
NMP	-0.73

The structures shown in Fig. 3 suggest that during the adsorption process, the sulfur or oxygen atoms in thiourea and urea can form a bond with the Pb^{2+} cation on the MAPbI_3 surface, which may explain the electron donation observed in experiments. This interaction between oxygen or sulfur and Pb^{2+} is also present in the solvent.

Figure S38: DMF, DMSO, NMP, urea and thiourea on the MAPbI_3 surface.

In conclusion, the higher polarity of thiourea enables it to form a strong bond with I^- , while its moderate adsorption energy allows it to desorb easily on the surface during annealing. These two factors make thiourea an ideal ion carrier, enhancing the mobility of I^- . Additionally, thiourea can improve the stability of MAPbI_3 by passivating the Pb^{2+} cations on the surface. While urea also forms a strong bond with I^- , its performance as an ion carrier is diminished due to its overly strong bond with Pb^{2+} . Therefore, as an additive, thiourea outperforms urea.

Figure S33: **a**, ^{207}Pb MAS NMR spectra of a physical mixture of $\text{MAPbI}_3 + \text{MAPbBr}_3$ and $\text{MAPbI}_3 + \text{MAPbBr}_3 + 10\% \text{MACl}$ before and after annealing at $80\text{ }^\circ\text{C}$ for 500 min. **b**, Corresponding normalized intensity of the formation of $[\text{PbI}_{6-x}\text{Br}_x]$ environments in mixed halide perovskite phases as a function of annealing time at $80\text{ }^\circ\text{C}$. Fits are based on the Johnson-Mehl-Avrami-Kolmogorov model and fit parameters are summarized in Table S3.

Table S3: Fitting parameters for the build-up curve corresponding to the phase formation of $\text{MAPbI}_{1.5}\text{Br}_{1.5}$ and $\text{MAPbI}_{1.5}\text{Br}_{1.5} + 10 \text{ mol-}\% \text{MACl}$ extracted from the formation of $[\text{PbI}_{6-x}\text{Br}_x]$ environments, as shown in Fig. S33.

starting sample composition	A	k / min^{-1}	n
$\text{MAPbBr}_3 + \text{MAPbI}_3$	1	$4.7 \cdot 10^{-4}$	0.32
$\text{MAPbBr}_3 + \text{MAPbI}_3 + 10 \text{ mol-}\% \text{MACl}$	1	$1.2 \cdot 10^{-3}$	0.30

We believe, that the additional insights presented, substantially strengthen the manuscript and open up new angles for process development. We are therefore convinced that the referee will find the manuscript suitable for publication now.